# Variational Discriminator Bottleneck:
## Improving Imitation Learning, Inverse RL, and GANs by Constraining Information Flow

**Xue Bin Peng & Angjoo Kanazawa & Sam Toyer & Pieter Abbeel & Sergey Levine**
University of California, Berkeley
{xbpeng,kanazawa,sdt,pabbeel,svlevine}@berkeley.edu

## Abstract

Adversarial learning methods have been proposed for a wide range of applications, but the training of adversarial models can be notoriously unstable. Effectively balancing the performance of the generator and discriminator is critical, since a discriminator that achieves very high accuracy will produce relatively uninformative gradients. In this work, we propose a simple and general technique to constrain information flow in the discriminator by means of an information bottleneck. By enforcing a constraint on the mutual information between the observations and the discriminator's internal representation, we can effectively modulate the discriminator's accuracy and maintain useful and informative gradients. We demonstrate that our proposed variational discriminator bottleneck (VDB) leads to significant improvements across three distinct application areas for adversarial learning algorithms. Our primary evaluation studies the applicability of the VDB to imitation learning of dynamic continuous control skills, such as running. We show that our method can learn such skills directly from *raw* video demonstrations, substantially outperforming prior adversarial imitation learning methods. The VDB can also be combined with adversarial inverse reinforcement learning to learn parsimonious reward functions that can be transferred and re-optimized in new settings. Finally, we demonstrate that VDB can train GANs more effectively for image generation, improving upon a number of prior stabilization methods. (Video[1])

## 1 Introduction

Adversarial learning methods provide a promising approach to modeling distributions over high-dimensional data with complex internal correlation structures. These methods generally use a discriminator to supervise the training of a generator in order to produce samples that are indistinguishable from the data. A particular instantiation is generative adversarial networks, which can be used for high-fidelity generation of images (Goodfellow et al., 2014; Karras et al., 2017) and other high-dimensional data (Vondrick et al., 2016; Xie et al., 2018; Donahue et al., 2018). Adversarial methods can also be used to learn reward functions in the framework of inverse reinforcement learning (Finn et al., 2016a; Fu et al., 2017), or to directly imitate demonstrations (Ho & Ermon, 2016). However, they suffer from major optimization challenges, one of which is balancing the performance of the generator and discriminator. A discriminator that achieves very high accuracy can produce relatively uninformative gradients, but a weak discriminator can also hamper the generator's ability to learn. These challenges have led to widespread interest in a variety of stabilization methods for adversarial learning algorithms (Arjovsky et al., 2017; Kodali et al., 2017; Berthelot et al., 2017).

In this work, we propose a simple regularization technique for adversarial learning, which constrains the information flow from the inputs to the discriminator using a variational approximation to the information bottleneck. By enforcing a constraint on the mutual information between the input observations and the discriminator's internal representation, we can encourage the discriminator to learn a representation that has heavy overlap between the data and the generator's distribution, thereby effectively modulating the discriminator's accuracy and maintaining useful and informative

---

[1]xbpeng.github.io/projects/VDB/

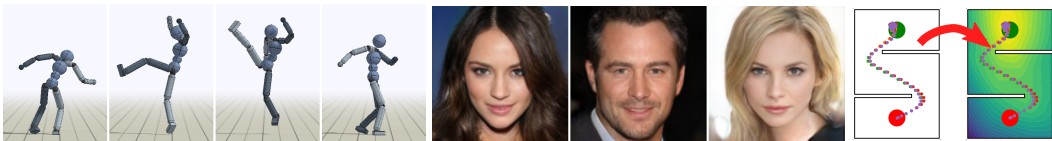

Figure 1: Our method is general and can be applied to a broad range of adversarial learning tasks. **Left:** Motion imitation with adversarial imitation learning. **Middle:** Image generation. **Right:** Learning transferable reward functions through adversarial inverse reinforcement learning.

gradients for the generator. Our approach to stabilizing adversarial learning can be viewed as an adaptive variant of instance noise (Salimans et al., 2016; Sønderby et al., 2016; Arjovsky & Bottou, 2017). However, we show that the adaptive nature of this method is critical. Constraining the mutual information between the discriminator's internal representation and the input allows the regularizer to directly limit the discriminator's accuracy, which automates the choice of noise magnitude and applies this noise to a compressed representation of the input that is specifically optimized to model the most discerning differences between the generator and data distributions.

The main contribution of this work is the variational discriminator bottleneck (VDB), an adaptive stochastic regularization method for adversarial learning that substantially improves performance across a range of different application domains, examples of which are available in Figure 1. Our method can be easily applied to a variety of tasks and architectures. First, we evaluate our method on a suite of challenging imitation tasks, including learning highly acrobatic skills from mocap data with a simulated humanoid character. Our method also enables characters to learn dynamic continuous control skills directly from raw video demonstrations, and drastically improves upon previous work that uses adversarial imitation learning. We further evaluate the effectiveness of the technique for inverse reinforcement learning, which recovers a reward function from demonstrations in order to train future policies. Finally, we apply our framework to image generation using generative adversarial networks, where employing VDB improves the performance in many cases.

## 2 RELATED WORK

Recent years have seen an explosion of adversarial learning techniques, spurred by the success of generative adversarial networks (GANs) (Goodfellow et al., 2014). A GAN framework is commonly composed of a discriminator and a generator, where the discriminator's objective is to classify samples as real or fake, while the generator's objective is to produce samples that fool the discriminator. Similar frameworks have also been proposed for inverse reinforcement learning (IRL) (Finn et al., 2016b) and imitation learning (Ho & Ermon, 2016). The training of adversarial models can be extremely unstable, with one of the most prevalent challenges being balancing the interplay between the discriminator and the generator (Berthelot et al., 2017). The discriminator can often overpower the generator, easily differentiating between real and fake samples, thus providing the generator with uninformative gradients for improvement (Che et al., 2016). Alternative loss functions have been proposed to mitigate this problem (Mao et al., 2016; Zhao et al., 2016; Arjovsky et al., 2017). Regularizers have been incorporated to improve stability and convergence, such as gradient penalties (Kodali et al., 2017; Gulrajani et al., 2017a; Mescheder et al., 2018), reconstruction loss (Che et al., 2016), and a myriad of other heuristics (Sønderby et al., 2016; Salimans et al., 2016; Arjovsky & Bottou, 2017; Berthelot et al., 2017). Task-specific architectural designs can also substantially improve performance (Radford et al., 2015; Karras et al., 2017). Similarly, our method also aims to regularize the discriminator in order to improve the feedback provided to the generator. But instead of explicit regularization of gradients or architecture-specific constraints, we apply a general information bottleneck to the discriminator, which previous works have shown to encourage networks to ignore irrelevant cues (Achille & Soatto, 2017). We hypothesize that this then allows the generator to focus on improving the most discerning differences between real and fake samples.

Adversarial techniques have also been applied to inverse reinforcement learning (Fu et al., 2017), where a reward function is recovered from demonstrations, which can then be used to train policies to reproduce a desired skill. Finn et al. (2016a) showed an equivalence between maximum entropy IRL and GANs. Similar techniques have been developed for adversarial imitation learning (Ho & Ermon, 2016; Merel et al., 2017), where agents learn to imitate demonstrations without explicitly recovering

a reward function. One advantage of adversarial methods is that by leveraging a discriminator in place of a reward function, they can be applied to imitate skills where reward functions can be difficult to engineer. However, the performance of policies trained through adversarial methods still falls short of those produced by manually designed reward functions, when such reward functions are available (Rajeswaran et al., 2017; Peng et al., 2018). We show that our method can significantly improve upon previous works that use adversarial techniques, and produces results of comparable quality to those from state-of-the-art approaches that utilize manually engineered reward functions.

Our variational discriminator bottleneck is based on the information bottleneck (Tishby & Zaslavsky, 2015), a technique for regularizing internal representations to minimize the mutual information with the input. Intuitively, a compressed representation can improve generalization by ignoring irrelevant distractors present in the original input. The information bottleneck can be instantiated in practical deep models by leveraging a variational bound and the reparameterization trick, inspired by a similar approach in variational autoencoders (VAE) (Kingma & Welling, 2013). The resulting *variational* information bottleneck approximates this compression effect in deep networks (Alemi et al., 2016; Achille & Soatto, 2017). A similar bottleneck has also been applied to learn disentangled representations (Higgins et al., 2017). Building on the success of VAEs and GANs, a number of efforts have been made to combine the two. Makhzani et al. (2016) used adversarial discriminators during the training of VAEs to encourage the marginal distribution of the latent encoding to be similar to the prior distribution, similar techniques include Mescheder et al. (2017) and Chen et al. (2018). Conversely, Larsen et al. (2016) modeled the generator of a GAN using a VAE. Zhao et al. (2016) used an autoencoder instead of a VAE to model the discriminator, but does not enforce an information bottleneck on the encoding. While instance noise is widely used in modern architectures (Salimans et al., 2016; Sønderby et al., 2016; Arjovsky & Bottou, 2017), we show that explicitly enforcing an information bottleneck leads to improved performance over simply adding noise for a variety of applications.

## 3 PRELIMINARIES

In this section, we provide a review of the variational information bottleneck proposed by Alemi et al. (2016) in the context of supervised learning. Our variational discriminator bottleneck is based on the same principle, and can be instantiated in the context of GANs, inverse RL, and imitation learning. Given a dataset $\{\mathbf{x}_i, \mathbf{y}_i\}$, with features $\mathbf{x}_i$ and labels $\mathbf{y}_i$, the standard maximum likelihood estimate $q(\mathbf{y}_i|\mathbf{x}_i)$ can be determined according to

$$\min_q \quad \mathbb{E}_{\mathbf{x},\mathbf{y}\sim p(\mathbf{x},\mathbf{y})}\left[-\log q(\mathbf{y}|\mathbf{x})\right]. \tag{1}$$

Unfortunately, this estimate is prone to overfitting, and the resulting model can often exploit idiosyncrasies in the data (Krizhevsky et al., 2012; Srivastava et al., 2014). Alemi et al. (2016) proposed regularizing the model using an information bottleneck to encourage the model to focus only on the most discriminative features. The bottleneck can be incorporated by first introducing an encoder $E(\mathbf{z}|\mathbf{x})$ that maps the features $\mathbf{x}$ to a latent distribution over $Z$, and then enforcing an upper bound $I_c$ on the mutual information between the encoding and the original features $I(X, Z)$. This results in the following regularized objective $J(q, E)$

$$J(q, E) = \min_{q,E} \quad \mathbb{E}_{\mathbf{x},\mathbf{y}\sim p(\mathbf{x},\mathbf{y})}\left[\mathbb{E}_{\mathbf{z}\sim E(\mathbf{z}|\mathbf{x})}\left[-\log q(\mathbf{y}|\mathbf{z})\right]\right]$$
$$\text{s.t.} \quad I(X, Z) \leq I_c. \tag{2}$$

Note that the model $q(\mathbf{y}|\mathbf{z})$ now maps samples from the latent distribution $\mathbf{z}$ to the label $\mathbf{y}$. The mutual information is defined according to

$$I(X, Z) = \int p(\mathbf{x}, \mathbf{z}) \log \frac{p(\mathbf{x}, \mathbf{z})}{p(\mathbf{x})p(\mathbf{z})} \, d\mathbf{x} \, d\mathbf{z} = \int p(\mathbf{x})E(\mathbf{z}|\mathbf{x}) \log \frac{E(\mathbf{z}|\mathbf{x})}{p(\mathbf{z})} \, d\mathbf{x} \, d\mathbf{z}, \tag{3}$$

where $p(\mathbf{x})$ is the distribution given by the dataset. Computing the marginal distribution $p(\mathbf{z}) = \int E(\mathbf{z}|\mathbf{x}) \, p(\mathbf{x}) \, d\mathbf{x}$ can be challenging. Instead, a variational lower bound can be obtained by using an approximation $r(\mathbf{z})$ of the marginal. Since $\text{KL}\left[p(\mathbf{z})||r(\mathbf{z})\right] \geq 0$, $\int p(\mathbf{z}) \log p(\mathbf{z}) \, d\mathbf{z} \geq \int p(\mathbf{z}) \log r(\mathbf{z}) \, d\mathbf{z}$, an upper bound on $I(X, Z)$ can be obtained via the KL divergence,

$$I(X, Z) \leq \int p(\mathbf{x})E(\mathbf{z}|\mathbf{x}) \log \frac{E(\mathbf{z}|\mathbf{x})}{r(\mathbf{z})} \, d\mathbf{x} \, d\mathbf{z} = \mathbb{E}_{\mathbf{x}\sim p(\mathbf{x})}\left[\text{KL}\left[E(\mathbf{z}|\mathbf{x})||r(\mathbf{z})\right]\right]. \tag{4}$$

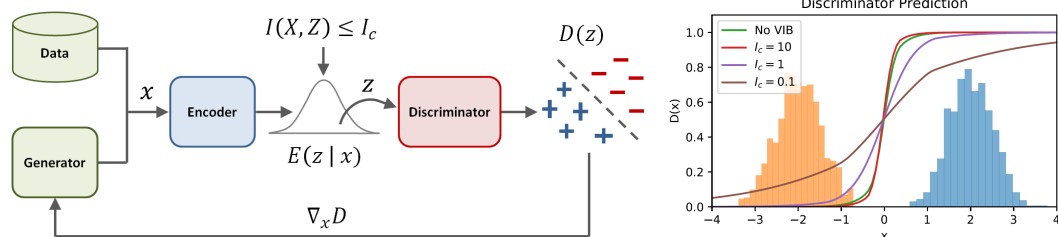

**Figure 2: Left:** Overview of the variational discriminator bottleneck. The encoder first maps samples $\mathbf{x}$ to a latent distribution $E(\mathbf{z}|\mathbf{x})$. The discriminator is then trained to classify samples $\mathbf{z}$ from the latent distribution. An information bottleneck $I(X, Z) \leq I_c$ is applied to $Z$. **Right:** Visualization of discriminators trained to differentiate two Gaussians with different KL bounds $I_c$.

This provides an upper bound on the regularized objective $\tilde{J}(q, E) \geq J(q, E)$,

$$\tilde{J}(q, E) = \min_{q, E} \quad \mathbb{E}_{\mathbf{x}, \mathbf{y} \sim p(\mathbf{x}, \mathbf{y})} \left[ \mathbb{E}_{\mathbf{z} \sim E(\mathbf{z}|\mathbf{x})} \left[ -\log q(\mathbf{y}|\mathbf{z}) \right] \right]$$
$$\text{s.t.} \quad \mathbb{E}_{\mathbf{x} \sim p(\mathbf{x})} \left[ \text{KL} \left[ E(\mathbf{z}|\mathbf{x}) || r(\mathbf{z}) \right] \right] \leq I_c. \tag{5}$$

To solve this problem, the constraint can be subsumed into the objective with a coefficient $\beta$

$$\min_{q, E} \quad \mathbb{E}_{\mathbf{x}, \mathbf{y} \sim p(\mathbf{x}, \mathbf{y})} \left[ \mathbb{E}_{\mathbf{z} \sim E(\mathbf{z}|\mathbf{x})} \left[ -\log q(\mathbf{y}|\mathbf{z}) \right] \right] + \beta \left( \mathbb{E}_{\mathbf{x} \sim p(\mathbf{x})} \left[ \text{KL} \left[ E(\mathbf{z}|\mathbf{x}) || r(\mathbf{z}) \right] - I_c \right) \right). \tag{6}$$

Alemi et al. (2016) evaluated the method on supervised learning tasks, and showed that models trained with a VIB can be less prone to overfitting and more robust to adversarial examples.

## 4 VARIATIONAL DISCRIMINATOR BOTTLENECK

To outline our method, we first consider a standard GAN framework consisting of a discriminator $D$ and a generator $G$, where the goal of the discriminator is to distinguish between samples from the target distribution $p^*(\mathbf{x})$ and samples from the generator $G(\mathbf{x})$,

$$\max_{G} \min_{D} \quad \mathbb{E}_{\mathbf{x} \sim p^*(\mathbf{x})} \left[ -\log \left( D(\mathbf{x}) \right) \right] + \mathbb{E}_{\mathbf{x} \sim G(\mathbf{x})} \left[ -\log \left( 1 - D(\mathbf{x}) \right) \right].$$

We incorporate a variational information bottleneck by introducing an encoder $E$ into the discriminator that maps a sample $\mathbf{x}$ to a stochastic encoding $\mathbf{z} \sim E(\mathbf{z}|\mathbf{x})$, and then apply a constraint $I_c$ on the mutual information $I(X, Z)$ between the original features and the encoding. $D$ is then trained to classify samples drawn from the encoder distribution. A schematic illustration of the framework is available in Figure 2. The regularized objective $J(D, E)$ for the discriminator is given by

$$J(D, E) = \min_{D, E} \quad \mathbb{E}_{x \sim p^*(\mathbf{x})} \left[ \mathbb{E}_{\mathbf{z} \sim E(\mathbf{z}|\mathbf{x})} \left[ -\log \left( D(\mathbf{z}) \right) \right] \right] + \mathbb{E}_{\mathbf{x} \sim G(\mathbf{x})} \left[ \mathbb{E}_{\mathbf{z} \sim E(\mathbf{z}|\mathbf{x})} \left[ -\log \left( 1 - D(\mathbf{z}) \right) \right] \right]$$
$$\text{s.t.} \quad \mathbb{E}_{\mathbf{x} \sim \tilde{p}(\mathbf{x})} \left[ \text{KL} \left[ E(\mathbf{z}|\mathbf{x}) || r(\mathbf{z}) \right] \right] \leq I_c, \tag{7}$$

with $\tilde{p} = \frac{1}{2} p^* + \frac{1}{2} G$ being a mixture of the target distribution and the generator. We refer to this regularizer as the variational discriminator bottleneck (VDB). To optimize this objective, we can introduce a Lagrange multiplier $\beta$,

$$J(D, E) = \min_{D, E} \max_{\beta \geq 0} \quad \mathbb{E}_{\mathbf{x} \sim p^*(\mathbf{x})} \left[ \mathbb{E}_{\mathbf{z} \sim E(\mathbf{z}|\mathbf{x})} \left[ -\log \left( D(\mathbf{z}) \right) \right] \right] + \mathbb{E}_{\mathbf{x} \sim G(\mathbf{x})} \left[ \mathbb{E}_{\mathbf{z} \sim E(\mathbf{z}|\mathbf{x})} \left[ -\log \left( 1 - D(\mathbf{z}) \right) \right] \right]$$
$$+ \beta \left( \mathbb{E}_{\mathbf{x} \sim \tilde{p}(\mathbf{x})} \left[ \text{KL} \left[ E(\mathbf{z}|\mathbf{x}) || r(\mathbf{z}) \right] \right] - I_c \right). \tag{8}$$

As we will discuss in Section 4.1 and demonstrate in our experiments, enforcing a specific mutual information budget between $\mathbf{x}$ and $\mathbf{z}$ is critical for good performance. We therefore adaptively update $\beta$ via dual gradient descent to enforce a specific constraint $I_c$ on the mutual information,

$$D, E \leftarrow \arg \min_{D, E} \mathcal{L}(D, E, \beta)$$
$$\beta \leftarrow \max \left( 0, \ \beta + \alpha_\beta \left( \mathbb{E}_{\mathbf{x} \sim \tilde{p}(\mathbf{x})} \left[ \text{KL} \left[ E(\mathbf{z}|\mathbf{x}) || r(\mathbf{z}) \right] - I_c \right) \right) \right), \tag{9}$$

where $\mathcal{L}(D, E, \beta)$ is the Lagrangian

$$\mathcal{L}(D, E, \beta) = \mathbb{E}_{\mathbf{x} \sim p^*(\mathbf{x})} \left[ \mathbb{E}_{\mathbf{z} \sim E(\mathbf{z}|\mathbf{x})} \left[ -\log\left( D(\mathbf{z}) \right) \right] \right] + \mathbb{E}_{\mathbf{x} \sim G(\mathbf{x})} \left[ \mathbb{E}_{\mathbf{z} \sim E(\mathbf{z}|\mathbf{x})} \left[ -\log\left( 1 - D(\mathbf{z}) \right) \right] \right]$$
$$+ \beta \left( \mathbb{E}_{\mathbf{x} \sim \tilde{p}(\mathbf{x})} \left[ \mathrm{KL}\left[ E(\mathbf{z}|\mathbf{x}) \| r(\mathbf{z}) \right] \right] - I_c \right) , \tag{10}$$

and $\alpha_\beta$ is the stepsize for the dual variable in dual gradient descent (Boyd & Vandenberghe, 2004). In practice, we perform only one gradient step on $D$ and $E$, followed by an update to $\beta$. We refer to a GAN that incorporates a VDB as a variational generative adversarial network (VGAN).

In our experiments, the prior $r(\mathbf{z}) = \mathcal{N}(0, I)$ is modeled with a standard Gaussian. The encoder $E(\mathbf{z}|\mathbf{x}) = \mathcal{N}(\mu_E(\mathbf{x}), \Sigma_E(\mathbf{x}))$ models a Gaussian distribution in the latent variables $Z$, with mean $\mu_E(\mathbf{x})$ and diagonal covariance matrix $\Sigma_E(\mathbf{x})$. When computing the KL loss, each batch of data contains an equal number of samples from $p^*(x)$ and $G(x)$. We use a simplified objective for the generator,

$$\max_G \quad \mathbb{E}_{\mathbf{x} \sim G(\mathbf{x})} \left[ -\log\left( 1 - D(\mu_E(\mathbf{x})) \right) \right] . \tag{11}$$

where the KL penalty is excluded from the generator's objective. Instead of computing the expectation over $Z$, we found that approximating the expectation by evaluating $D$ at the mean $\mu_E(\mathbf{x})$ of the encoder's distribution was sufficient for our tasks. The discriminator is modeled with a single linear unit followed by a sigmoid $D(\mathbf{z}) = \sigma(\mathbf{w}_D^T \mathbf{z} + \mathbf{b}_D)$, with weights $\mathbf{w}_D$ and bias $\mathbf{b}_D$.

## 4.1 Discussion and Analysis

To interpret the effects of the VDB, we consider the results presented by Arjovsky & Bottou (2017), which show that for two distributions with disjoint support, the optimal discriminator can perfectly classify all samples and its gradients will be zero almost everywhere. Thus, as the discriminator converges to the optimum, the gradients for the generator vanishes accordingly. To address this issue, Arjovsky & Bottou (2017) proposed applying continuous noise to the discriminator inputs, thereby ensuring that the distributions have continuous support everywhere. In practice, if the original distributions are sufficiently distant from each other, the added noise will have negligible effects. As shown by Mescheder et al. (2017), the optimal choice for the variance of the noise to ensure convergence can be quite delicate. In our method, by first using a learned encoder to map the inputs to an embedding and then applying an information bottleneck on the embedding, we can dynamically adjust the variance of the noise such that the distributions not only share support in the embedding space, but also have significant overlap. Since the minimum amount of information required for binary classification is 1 bit, by selecting an information constraint $I_c < 1$, the discriminator is prevented from from perfectly differentiating between the distributions. To illustrate the effects of the VDB, we consider a simple task of training a discriminator to differentiate between two Gaussian distributions. Figure 2 visualizes the decision boundaries learned with different bounds $I_c$ on the mutual information. Without a VDB, the discriminator learns a sharp decision boundary, resulting in vanishing gradients for much of the space. But as $I_c$ decreases and the bound tightens, the decision boundary is smoothed, providing more informative gradients that can be leveraged by the generator.

Taking this analysis further, we can extend Theorem 3.2 from Arjovsky & Bottou (2017) to analyze the VDB, and show that the gradient of the generator will be non-degenerate for a small enough constraint $I_c$, under some additional simplifying assumptions. The result in Arjovsky & Bottou (2017) states that the gradient consists of vectors that point toward samples on the data manifold, multiplied by coefficients that depend on the noise. However, these coefficients may be arbitrarily small if the generated samples are far from real samples, and the noise is not large enough. This can still cause the generator gradient to vanish. In the case of the VDB, the constraint ensures that these coefficients are always bounded below. Due to space constraints, this result is presented in Appendix A.

## 4.2 VAIL: Variational Adversarial Imitation Learning

To extend the VDB to imitation learning, we start with the generative adversarial imitation learning (GAIL) framework (Ho & Ermon, 2016), where the discriminator's objective is to differentiate between the state distribution induced by a target policy $\pi^*(\mathbf{s})$ and the state distribution of the agent's policy $\pi(\mathbf{s})$,

$$\max_\pi \min_D \quad \mathbb{E}_{\mathbf{s} \sim \pi^*(\mathbf{s})} \left[ -\log\left( D(\mathbf{s}) \right) \right] + \mathbb{E}_{\mathbf{s} \sim \pi(\mathbf{s})} \left[ -\log\left( 1 - D(\mathbf{s}) \right) \right] .$$

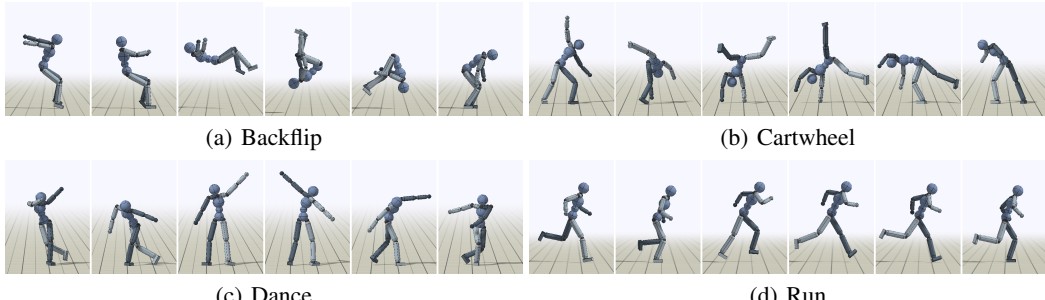

(a) Backflip  (b) Cartwheel

(c) Dance  (d) Run

Figure 3: Simulated humanoid performing various skills. VAIL is able to closely imitate a broad range of skills from mocap data.

The discriminator is trained to maximize the likelihood assigned to states from the target policy, while minimizing the likelihood assigned to states from the agent's policy. The discriminator also serves as the reward function for the agent, which encourages the policy to visit states that, to the discriminator, appear indistinguishable from the demonstrations. Similar to the GAN framework, we can incorporate a VDB into the discriminator,

$$J(D, E) = \min_{D, E} \max_{\beta \geq 0} \mathbb{E}_{\mathbf{s} \sim \pi^*(\mathbf{s})} \left[ \mathbb{E}_{\mathbf{z} \sim E(\mathbf{z}|\mathbf{s})} \left[ -\log \left( D(\mathbf{z}) \right) \right] \right] + \mathbb{E}_{\mathbf{s} \sim \pi(\mathbf{s})} \left[ \mathbb{E}_{\mathbf{z} \sim E(\mathbf{z}|\mathbf{s})} \left[ -\log \left( 1 - D(\mathbf{z}) \right) \right] \right]$$

$$+ \beta \left( \mathbb{E}_{\mathbf{s} \sim \tilde{\pi}(\mathbf{s})} \left[ \mathrm{KL} \left[ E(\mathbf{z}|\mathbf{s}) || r(\mathbf{z}) \right] \right] - I_c \right). \tag{12}$$

where $\tilde{\pi} = \frac{1}{2}\pi^* + \frac{1}{2}\pi$ represents a mixture of the target policy and the agent's policy. The reward for $\pi$ is then specified by the discriminator $r_t = -\log\left(1 - D(\mu_E(\mathbf{s}))\right)$. We refer to this method as variational adversarial imitation learning (VAIL).

## 4.3 VAIRL: VARIATIONAL ADVERSARIAL INVERSE REINFORCEMENT LEARNING

The VDB can also be applied to adversarial inverse reinforcement learning (Fu et al., 2017) to yield a new algorithm which we call variational adversarial inverse reinforcement learning (VAIRL). AIRL operates in a similar manner to GAIL, but with a discriminator of the form

$$D(\mathbf{s}, \mathbf{a}, \mathbf{s}') = \frac{\exp\left(f(\mathbf{s}, \mathbf{a}, \mathbf{s}')\right)}{\exp\left(f(\mathbf{s}, \mathbf{a}, \mathbf{s}')\right) + \pi(\mathbf{a}|\mathbf{s})}, \tag{13}$$

where $f(\mathbf{s}, \mathbf{a}, \mathbf{s}') = g(\mathbf{s}, \mathbf{a}) + \gamma h(\mathbf{s}') - h(\mathbf{s})$, with $g$ and $h$ being learned functions. Under certain restrictions on the environment, Fu et al. show that if $g(\mathbf{s}, \mathbf{a})$ is defined to depend only on the current state $\mathbf{s}$, the optimal $g(\mathbf{s})$ recovers the expert's true reward function $r^*(\mathbf{s})$ up to a constant $g^*(\mathbf{s}) = r^*(\mathbf{s}) + \mathrm{const}$. In this case, the learned reward can be re-used to train policies in environments with different dynamics, and will yield the same policy as if the policy was trained under the expert's true reward. In contrast, GAIL's discriminator typically cannot be re-optimized in this way (Fu et al., 2017). In VAIRL, we introduce stochastic encoders $E_g(\mathbf{z}_g|\mathbf{s})$, $E_h(\mathbf{z}_h|\mathbf{s})$, and $g(\mathbf{z}_g)$, $h(\mathbf{z}_h)$ are modified to be functions of the encoding. We can reformulate Equation 13 as

$$D(\mathbf{s}, \mathbf{a}, \mathbf{z}) = \frac{\exp\left(f(\mathbf{z}_g, \mathbf{z}_h, \mathbf{z}'_h)\right)}{\exp\left(f(\mathbf{z}_g, \mathbf{z}_h, \mathbf{z}'_h)\right) + \pi(\mathbf{a}|\mathbf{s})},$$

for $\mathbf{z} = (\mathbf{z}_g, \mathbf{z}_h, \mathbf{z}'_h)$ and $f(\mathbf{z}_g, \mathbf{z}_h, \mathbf{z}'_h) = D_g(\mathbf{z}_g) + \gamma D_h(\mathbf{z}'_h) - D_h(\mathbf{z}_h)$. We then obtain a modified objective of the form

$$J(D, E) = \min_{D, E} \max_{\beta \geq 0} \ \mathbb{E}_{\mathbf{s}, \mathbf{s}' \sim \pi^*(\mathbf{s}, \mathbf{s}')} \left[ \mathbb{E}_{\mathbf{z} \sim E(\mathbf{z}|\mathbf{s}, \mathbf{s}')} \left[ -\log\left( D(\mathbf{s}, \mathbf{a}, \mathbf{z}) \right) \right] \right]$$

$$+ \mathbb{E}_{\mathbf{s}, \mathbf{s}' \sim \pi(\mathbf{s}, \mathbf{s}')} \left[ \mathbb{E}_{\mathbf{z} \sim E(\mathbf{z}|\mathbf{s}, \mathbf{s}')} \left[ -\log\left( 1 - D(\mathbf{s}, \mathbf{a}, \mathbf{z}) \right) \right] \right]$$

$$+ \beta \left( \mathbb{E}_{\mathbf{s}, \mathbf{s}' \sim \tilde{\pi}(\mathbf{s}, \mathbf{s}')} \left[ \mathrm{KL} \left[ E(\mathbf{z}|\mathbf{s}, \mathbf{s}') || r(\mathbf{z}) \right] \right] - I_c \right),$$

where $\pi(s, s')$ denotes the joint distribution of successive states from a policy, and $E(\mathbf{z}|\mathbf{s}, \mathbf{s}') = E_g(\mathbf{z}_g|\mathbf{s}) \cdot E_h(\mathbf{z}_h|\mathbf{s}) \cdot E_h(\mathbf{z}'_h|\mathbf{s}')$.

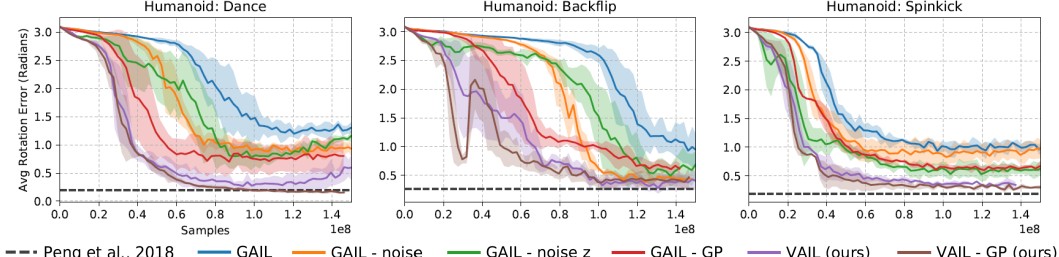

Figure 4: Learning curves comparing VAIL to other methods for motion imitation. Performance is measured using the average joint rotation error between the simulated character and the reference motion. Each method is evaluated with 3 random seeds.

| Method | Backflip | Cartwheel | Dance | Run | Spinkick |
|---|---|---|---|---|---|
| BC | 3.01 | 2.88 | 2.93 | 2.63 | 2.88 |
| Merel et al., 2017 | $1.33 \pm 0.03$ | $1.47 \pm 0.12$ | $2.61 \pm 0.30$ | $0.52 \pm 0.04$ | $1.82 \pm 0.35$ |
| GAIL | $0.74 \pm 0.15$ | $0.84 \pm 0.05$ | $1.31 \pm 0.16$ | $0.17 \pm 0.03$ | $1.07 \pm 0.03$ |
| GAIL - noise | $0.42 \pm 0.02$ | $0.92 \pm 0.07$ | $0.96 \pm 0.08$ | $0.21 \pm 0.05$ | $0.95 \pm 0.14$ |
| GAIL - noise z | $0.67 \pm 0.12$ | $0.72 \pm 0.04$ | $1.14 \pm 0.08$ | $0.14 \pm 0.03$ | $0.64 \pm 0.09$ |
| GAIL - GP | $0.62 \pm 0.09$ | $0.69 \pm 0.05$ | $0.80 \pm 0.32$ | $0.12 \pm 0.02$ | $0.64 \pm 0.04$ |
| VAIL (ours) | $\mathbf{0.36 \pm 0.13}$ | $0.40 \pm 0.08$ | $0.40 \pm 0.21$ | $0.13 \pm 0.01$ | $0.34 \pm 0.05$ |
| VAIL - GP (ours) | $0.46 \pm 0.17$ | $\mathbf{0.31 \pm 0.02}$ | $\mathbf{0.15 \pm 0.01}$ | $\mathbf{0.10 \pm 0.01}$ | $\mathbf{0.31 \pm 0.02}$ |
| Peng et al., 2018 | 0.26 | 0.21 | 0.20 | 0.14 | 0.19 |

Table 1: Average joint rotation error (radians) on humanoid motion imitation tasks. VAIL outperforms the other methods for all skills evaluated, except for policies trained using the manually-designed reward function from (Peng et al., 2018).

## 5 EXPERIMENTS

We evaluate our method on adversarial learning problems in imitation learning, inverse reinforcement learning, and image generation. In the case of imitation learning, we show that the VDB enables agents to learn complex motion skills from a single demonstration, including visual demonstrations provided in the form of video clips. We also show that the VDB improves the performance of inverse RL methods. Inverse RL aims to reconstruct a reward function from a set demonstrations, which can then used to perform the task in new environments, in contrast to imitation learning, which aims to recover a policy directly. Our method is also not limited to control tasks, and we demonstrate its effectiveness for unconditional image generation.

### 5.1 VAIL: VARIATIONAL ADVERSARIAL IMITATION LEARNING

The goal of the motion imitation tasks is to train a simulated character to mimic demonstrations provided by mocap clips recorded from human actors. Each mocap clip provides a sequence of target states $\{s_0^*, s_1^*, ..., s_T^*\}$ that the character should track at each timestep. We use a similar experimental setup as Peng et al. (2018), with a 34 degrees-of-freedom humanoid character. We found that the discriminator architecture can greatly affect the performance on complex skills. The particular architecture we employ differs substantially from those used in prior work (Merel et al., 2017), details of which are available in Appendix C. The encoding $Z$ is 128D and an information constraint of $I_c = 0.5$ is applied for all skills, with a dual stepsize of $\alpha_\beta = 10^{-5}$. All policies are trained using PPO (Schulman et al., 2017).

The motions learned by the policies are best seen in the supplementary video. Snapshots of the character's motions are shown in Figure 3. Each skill is learned from a single demonstration. VAIL is able to closely reproduce a variety of skills, including those that involve highly dynamics flips and complex contacts. We compare VAIL to a number of other techniques, including state-only GAIL (Ho & Ermon, 2016), GAIL with instance noise applied to the discriminator inputs (GAIL - noise), GAIL with instance noise applied to the last hidden layer (GAIL - noise z), and GAIL with a gradient penalty applied to the discriminator (GAIL - GP) (Mescheder et al., 2018). Since the VDB helps to prevent vanishing gradients, while GP mitigates exploding gradients, the two techniques can be seen as being complementary. Therefore, we also train a model that combines both VAIL and GP (VAIL -

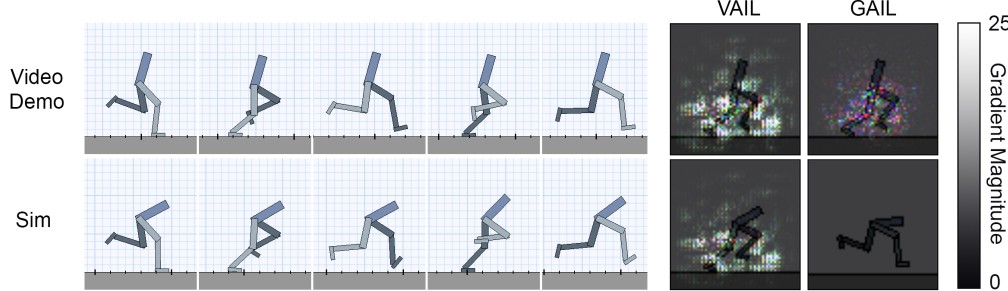

Figure 5: **Left:** Snapshots of the video demonstration and the simulated character trained with VAIL. The policy learns to run by directly imitating the video. **Right:** Saliency maps that visualize the magnitude of the discriminator's gradient with respect to all channels of the RGB input images from both the demonstration and the simulation. Pixel values are normalized between $[0, 1]$.

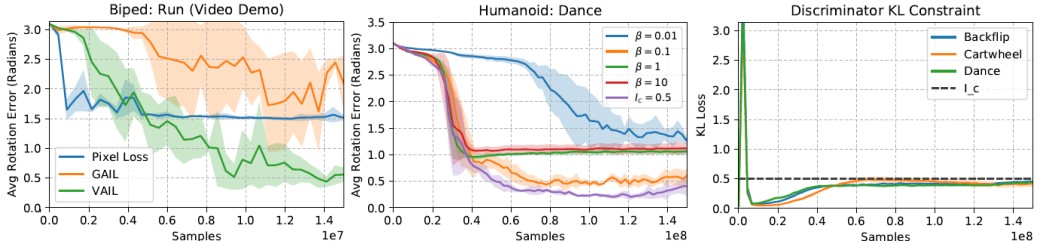

Figure 6: **Left:** Learning curves comparing policies for the video imitation task trained using a pixel-wise loss as the reward, GAIL, and VAIL. Only VAIL successfully learns to run from a video demonstration. **Middle:** Effect of training with fixed values of $\beta$ and adaptive $\beta$ ($I_c = 0.5$). **Right:**. KL loss over the course of training with adaptive $\beta$. The dual gradient descent update for $\beta$ effectively enforces the VDB constraint $I_c$.

GP). Implementation details for combining the VDB and GP are available in Appendix B. Learning curves for the various methods are shown in Figure 10 and Table 1 summarizes the performance of the final policies. Performance is measured in terms of the average joint rotation error between the simulated character and the reference motion. We also include a reimplementation of the method described by Merel et al. (2017). For the purpose of our experiments, GAIL denotes policies trained using our particular architecture but without a VDB, and Merel et al. (2017) denotes policies trained using an architecture that closely mirror those from previous work. Furthermore, we include comparisons to policies trained using the handcrafted reward from Peng et al. (2018), as well as policies trained via behavioral cloning (BC). Since mocap data does not provide expert actions, we use the policies from Peng et al. (2018) as oracles to provide state-action demonstrations, which are then used to train the BC policies via supervised learning. Each BC policy is trained with 10k samples from the oracle policies, while all other policies are trained from just a single demonstration, the equivalent of approximately 100 samples.

VAIL consistently outperforms previous adversarial methods, and VAIL - GP achieves the best performance overall. Simply adding instance noise to the inputs (Salimans et al., 2016) or hidden layer without the KL constraint (Sønderby et al., 2016) leads to worse performance, since the network can learn a latent representation that renders the effects of the noise negligible. Though training with the handcrafted reward still outperforms the adversarial methods, VAIL demonstrates comparable performance to the handcrafted reward without manual reward or feature engineering, and produces motions that closely resemble the original demonstrations. The method from Merel et al. (2017) was able to imitate simple skills such as running, but was unable to reproduce more acrobatic skills such as the backflip and spinkick. In the case of running, our implementation produces more natural gaits than the results reported in Merel et al. (2017). Behavioral cloning is unable to reproduce any of the skills, despite being provided with substantially more demonstration data than the other methods.

**Video Imitation:** While our method achieves substantially better results on motion imitation when compared to prior work, previous methods can still produce reasonable behaviors. However, if the demonstrations are provided in terms of the raw pixels from video clips, instead of mocap data, the imitation task becomes substantially harder. The goal of the agent is therefore to directly im-

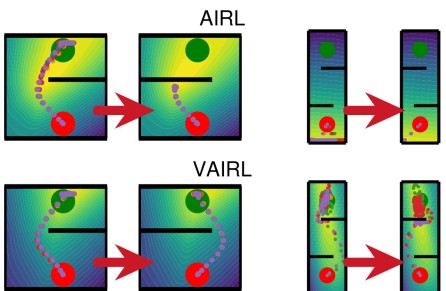

| Method | Transfer environments | |
|---|---|---|
| | C-maze | S-maze |
| GAIL | -24.6±7.2 | 1.0±1.3 |
| VAIL | -65.6±18.9 | 20.8±39.7 |
| AIRL | -15.3±7.8 | -0.2±0.1 |
| AIRL - GP | **-9.14±0.4** | -0.14±0.3 |
| VAIRL ($\beta = 0$) | -25.5±7.2 | 62.3±33.2 |
| VAIRL (ours) | -10.0±2.2 | 74.0±38.7 |
| VAIRL - GP (ours) | -9.18±0.4 | **156.5±5.6** |
| TRPO expert | -5.1 | 153.2 |

Figure 7: **Left:** C-Maze and S-Maze. When trained on the training maze on the left, AIRL learns a reward that overfits to the training task, and which cannot be transferred to the mirrored maze on the right. In contrast, VAIRL learns a smoother reward function that enables more-reliable transfer. **Right:** Performance on flipped test versions of our two training mazes. We report mean return (± std. dev.) over five runs, and the mean return for the expert used to generate demonstrations.

itate the skill depicted in the video. This is also a setting where manually engineering rewards is impractical, since simple losses like pixel distance do not provide a semantically meaningful measure of similarity. Figure 6 compares learning curves of policies trained with VAIL, GAIL, and policies trained using a reward function defined by the average pixel-wise difference between the frame $M_t^*$ from the video demonstration and a rendered image $M_t$ of the agent at each timestep $t$, $r_t = 1 - \frac{1}{3 \times 64^2}||M_t^* - M_t||^2$. Each frame is represented by a $64 \times 64$ RGB image.

Both GAIL and the pixel-loss are unable to learn the running gait. VAIL is the only method that successfully learns to imitate the skill from the video demonstration. Snapshots of the video demonstration and the simulated motion is available in Figure 5. To further investigate the effects of the VDB, we visualize the gradient of the discriminator with respect to images from the video demonstration and simulation. Saliency maps for discriminators trained with VAIL and GAIL are available in Figure 5. The VAIL discriminator learns to attend to spatially coherent image patches around the character, while the GAIL discriminator exhibits less structure. The magnitude of the gradients from VAIL also tend to be significantly larger than those from GAIL, which may suggests that VAIL is able to mitigate the problem of vanishing gradients present in GAIL.

**Adaptive Constraint:** To evaluate the effects of the adaptive $\beta$ updates, we compare policies trained with different fixed values of $\beta$ and policies where $\beta$ is updated adaptively to enforce a desired information constraint $I_c = 0.5$. Figure 6 illustrates the learning curves and the KL loss over the course of training. When $\beta$ is too small, performance reverts to that achieved by GAIL. Large values of $\beta$ help to smooth the discriminator landscape and improve learning speed during the early stages of training, but converges to a worse performance. Policies trained using dual gradient descent to adaptively update $\beta$ consistently achieves the best performance overall.

## 5.2 VAIRL: VARIATIONAL ADVERSARIAL INVERSE REINFORCEMENT LEARNING

Next, we use VAIRL to recover reward functions from demonstrations. Unlike the discriminator learned by VAIL, the reward function recovered by VAIRL can be re-optimized to train new policies from scratch in the same environment. In some cases, it can also be used to transfer similar behaviour to different environments. In Figure 7, we show the results of applying VAIRL to the C-maze from Fu et al. (2017), and a more complex S-maze; the simple 2D observation spaces of these tasks make it easy to interpret the recovered reward functions. In both mazes, the expert is trained to navigate from a start position at the bottom of the maze to a fixed target position at the top. We use each method to obtain an imitation policy and to approximate the expert's reward on the original maze. The recovered reward is then used to train a new policy to solve a left–right flipped version of the training maze. On the C-maze, we found that plain AIRL—without a gradient penalty— would sometimes overfit and fail to transfer to the new environment, as evidenced by the reward visualization in Figure 7 (left) and the higher return variance in Figure 7 (right). In contrast, by incorporating a VDB into AIRL, VAIRL learns a substantially smoother reward function that is more suitable for transfer. Furthermore, we found that in the S-maze with two internal walls, AIRL was too unstable to acquire a meaningful reward function. This was true even with the use of a gradient penalty. In contrast, VAIRL was able to learn a reasonable reward in most cases without a

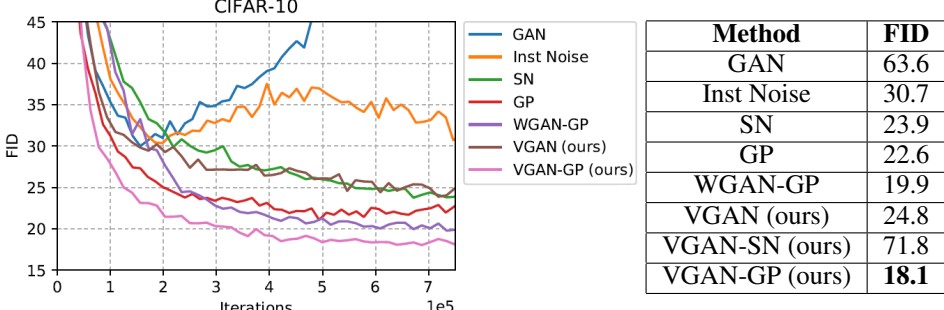

| Method | FID |
|---|---|
| GAN | 63.6 |
| Inst Noise | 30.7 |
| SN | 23.9 |
| GP | 22.6 |
| WGAN-GP | 19.9 |
| VGAN (ours) | 24.8 |
| VGAN-SN (ours) | 71.8 |
| VGAN-GP (ours) | **18.1** |

Figure 8: Comparison of VGAN and other methods on CIFAR-10, with performance evaluated using the Fréchet Inception Distance (FID).

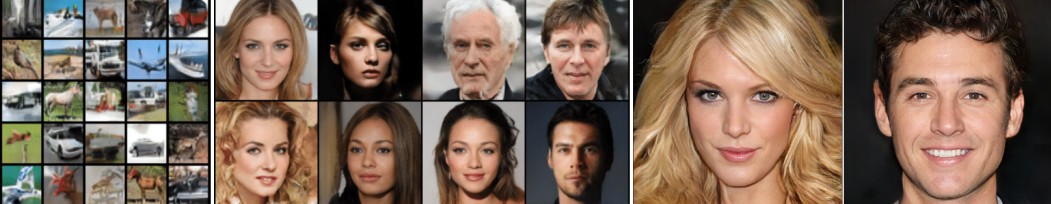

Figure 9: VGAN samples on CIFAR-10, CelebA 128×128, and CelebAHQ 1024×1024.

gradient penalty, and its performance improved even further with the addition of a gradient penalty. To evaluate the effects of the VDB, we observe that the performance of VAIRL drops on both tasks when the KL constraint is disabled ($\beta = 0$), suggesting that the improvements from the VDB cannot be attributed entirely to the noise introduced by the sampling process for **z**. Further details of these experiments and illustrations of the recovered reward functions are available in Appendix D.

### 5.3 VGAN: VARIATIONAL GENERATIVE ADVERSARIAL NETWORKS

Finally, we apply the VDB to image generation with generative adversarial networks, which we refer to as VGAN. Experiment are conducted on CIFAR-10 (Krizhevsky et al.), CelebA (Liu et al. (2015)), and CelebAHQ (Karras et al., 2018) datasets. We compare our approach to recent stabilization techniques: WGAN-GP (Gulrajani et al., 2017b), instance noise (Sønderby et al., 2016; Arjovsky & Bottou, 2017), spectral normalization (SN) (Miyato et al., 2018), and gradient penalty (GP) (Mescheder et al., 2018), as well as the original GAN (Goodfellow et al., 2014) on CIFAR-10. To measure performance, we report the Fréchet Inception Distance (FID) (Heusel et al., 2017), which has been shown to be more consistent with human evaluation. All methods are implemented using the same base model, built on the resnet architecture of Mescheder et al. (2018). Aside from tuning the KL constraint $I_c$ for VGAN, no additional hyperparameter optimization was performed to modify the settings provided by Mescheder et al. (2018). The performance of the various methods on CIFAR-10 are shown in Figure 8. While vanilla GAN and instance noise are prone to diverging as training progresses, VGAN remains stable. Note that instance noise can be seen as a non-adaptive version of VGAN without constraints on $I_c$. This experiment again highlights that there is a significant improvement from imposing the information bottleneck over simply adding instance noise. Combining both VDB and gradient penalty (VGAN - GP) achieves the best performance overall with an FID of 18.1. We also experimented with combining the VDB with SN, but this combination is prone to diverging. See Figure 9 for samples of images generated with our approach. Please refer to Appendix E for experimental details and more results.

## 6 CONCLUSION

We present the variational discriminator bottleneck, a general regularization technique for adversarial learning. Our experiments show that the VDB is broadly applicable to a variety of domains, and yields significant improvements over previous techniques on a number of challenging tasks. While our experiments have produced promising results for video imitation, the results have been primarily with videos of synthetic scenes. We believe that extending the technique to imitating real-world videos is an exciting direction. Another exciting direction for future work is a more in-depth theoretical analysis of the method, to derive convergence and stability results or conditions.

ACKNOWLEDGEMENTS

We would like to thank the anonymous reviewers for their helpful feedback, and AWS and NVIDIA for providing computational resources. This research was funded by an NSERC Postgraduate Scholarship, a Berkeley Fellowship for Graduate Study, BAIR, Huawei, and ONR PECASE N000141612723.

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

SUPPLEMENTARY MATERIAL

## A  ANALYSIS AND PROOFS

In this appendix, we show that the gradient of the generator when the discriminator is augmented with the VDB is non-degenerate, under some mild additional assumptions. First, we assume a pointwise constraint of the form $\mathrm{KL}[E(\mathbf{z}|\mathbf{x})\|r(\mathbf{z})] \leq I_c$ for all $\mathbf{x}$. In reality, we use an average KL constraint, since we found it to be more convenient to optimize, though a pointwise constraint is also possible to enforce by using the largest constraint violation to increment $\beta$. We could likely also extend the analysis to the average constraint, though we leave this to future work. The main theorem can then be stated as follows:

**Theorem A.1.** *Let $g(\mathbf{u})$ denote the generator's mapping from a noise vector $\mathbf{u} \sim p(\mathbf{u})$ to a point in $X$. Given the generator distribution $G(\mathbf{x})$ and data distribution $p^*(\mathbf{x})$, a VDB with an encoder $E(\mathbf{z}|\mathbf{x}) = \mathcal{N}(\mu_E(\mathbf{x}), \Sigma)$, and $\mathrm{KL}[E(\mathbf{z}|\mathbf{x})\|r(\mathbf{z})] \leq I_c$, the gradient passed to the generator has the form*

$$
\nabla_g \mathbb{E}_{\mathbf{u} \sim p(\mathbf{u})} \left[ \log \left( 1 - D^*(\mu_E(g(\mathbf{u}))) \right) \right]
$$
$$
= \mathbb{E}_{\mathbf{u} \sim p(\mathbf{u})} \Bigg[ a(\mathbf{u}) \int E(\mu_E(g(\mathbf{u}))|\mathbf{x}) \nabla_g \|\mu_E(g(\mathbf{u})) - \mu_E(\mathbf{x})\|^2 dp^*(\mathbf{x})
$$
$$
- b(\mathbf{u}) \int E(\mu_E(g(\mathbf{u}))|\mathbf{x}) \nabla_g \|\mu_E(g(\mathbf{u})) - \mu_E(\mathbf{x})\|^2 dG(\mathbf{x}) \Bigg]
$$

*where $D^*(\mathbf{z})$ is the optimal discriminator, $a(\mathbf{x})$ and $b(\mathbf{x})$ are positive functions, and we always have $E(\mu_E(g(\mathbf{u}))|\mathbf{x}) > C(I_c)$, where $C(I_c)$ is a continuous monotonic function, and $C(I_c) \to \delta > 0$ as $I_c \to 0$.*

Analysis for an encoder with an input-dependent variance $\Sigma(\mathbf{x})$ is also possible, but more involved. We'll further assume below for notational simplicity that $\Sigma$ is diagonal with diagonal values $\sigma^2$. This assumption is not required, but substantially simplifies the linear algebra. Analogously to Theorem 3.2 from Arjovsky & Bottou (2017), this theorem states that the gradient of the generator points in the direction of points in the data distribution, and away from points in the generator distribution. However, going beyond the theorem in Arjovsky & Bottou (2017), this result states that the coefficients on these vectors, given by $E(\mu_E(g(\mathbf{u}))|\mathbf{x})$, are always bounded below by a value that approaches a positive constant $\delta$ as we decrease $I_c$, meaning that the gradient does not vanish. The proof of the first part of this theorem is essentially identical to the proof presented by Arjovsky & Bottou (2017), but accounting for the fact that the noise is now injected into the latent space of the VDB, rather than being added directly to $\mathbf{x}$. This result assumes that $E(\mathbf{z}|\mathbf{x})$ has a learned but input-independent variance $\Sigma = \sigma^2 I$, though the proof can be repeated for an input-dependent or non-diagonal $\Sigma$:

*Proof.* Overloading $p^*(\mathbf{x})$ and $G(\mathbf{x})$, let $p^*(\mathbf{z})$ and $G(\mathbf{z})$ be the distribution of embeddings $\mathbf{z}$ under the real data and generator respectively. $p^*(\mathbf{z})$ is then given by

$$
p^*(\mathbf{z}) = \mathbb{E}_{\mathbf{x} \sim p^*(\mathbf{x})} [E(\mathbf{z}|\mathbf{x})] = \int E(\mathbf{z}|\mathbf{x}) dp^*(\mathbf{x}),
$$

and similarly for $G(z)$

$$
G(\mathbf{z}) = \mathbb{E}_{\mathbf{x} \sim G(\mathbf{x})} [E(\mathbf{z}|\mathbf{x})] = \int E(\mathbf{z}|\mathbf{x}) dG(\mathbf{x}),
$$

From Arjovsky & Bottou (2017), the optimal discriminator between $p^*(\mathbf{z})$ and $G(\mathbf{z})$ is

$$
D^*(\mathbf{z}) = \frac{p^*(\mathbf{z})}{p^*(\mathbf{z}) + G(\mathbf{z})}
$$

The gradient passed to the generator then has the form

$$\nabla_g \mathbb{E}_{\mathbf{u} \sim p(\mathbf{u})} \left[ \log\left(1 - D^*(\mu_E(g(\mathbf{u})))\right) \right]$$
$$= \mathbb{E}_{\mathbf{u} \sim p(\mathbf{u})} \left[ \nabla_g \log\left(G(\mu_E(g(\mathbf{u})))\right) - \nabla_g \log\left(p^*(\mu_E(g(\mathbf{u}))) + G(\mu_E(g(\mathbf{u})))\right) \right]$$
$$= \mathbb{E}_{\mathbf{u} \sim p(\mathbf{u})} \left[ \frac{\nabla_g G(\mu_E(g(\mathbf{u})))}{G(\mu_E(g(\mathbf{u})))} - \frac{\nabla_g p^*(\mu_E(g(\mathbf{u}))) + \nabla_g G(\mu_E(g(\mathbf{u})))}{p^*(\mu_E(g(\mathbf{u}))) + G(\mu_E(g(\mathbf{u})))} \right]$$
$$= \mathbb{E}_{\mathbf{u} \sim p(\mathbf{u})} \left[ \frac{1}{p^*(\mu_E(g(\mathbf{u}))) + G(\mu_E(g(\mathbf{u})))} \nabla_g \left[ -p^*(\mu_E(g(\mathbf{u}))) \right] \right.$$
$$\left. - \frac{1}{p^*(\mu_E(g(\mathbf{u}))) + G(\mu_E(g(\mathbf{u})))} \frac{p^*(\mu_E(g(\mathbf{u})))}{G(\mu_E(g(\mathbf{u})))} \nabla_g \left[ -G(\mu_E(g(\mathbf{u}))) \right] \right].$$

Let

$$a(\mathbf{u}) = \frac{1}{2\sigma^2} \frac{1}{p^*(\mu_E(g(\mathbf{u}))) + G(\mu_E(g(\mathbf{u})))}$$
$$b(\mathbf{u}) = \frac{1}{2\sigma^2} \frac{1}{p^*(\mu_E(g(\mathbf{u}))) + G(\mu_E(g(\mathbf{u})))} \frac{p^*(\mu_E(g(\mathbf{u})))}{G(\mu_E(g(\mathbf{u})))}.$$

We then have

$$\nabla_g \mathbb{E}_{\mathbf{u} \sim p(\mathbf{u})} \left[ \log\left(1 - D^*(\mu_E(g(\mathbf{u})))\right) \right]$$
$$= \mathbb{E}_{\mathbf{u} \sim p(\mathbf{u})} \left[ 2\sigma^2 \, a(\mathbf{u}) \nabla_g \left[ -p^*(\mu_E(g(\mathbf{u}))) \right] - 2\sigma^2 \, b(\mathbf{u}) \nabla_g \left[ -G(\mu_E(g(\mathbf{u}))) \right] \right]$$
$$= \mathbb{E}_{\mathbf{u} \sim p(\mathbf{u})} \left[ 2\sigma^2 \, a(\mathbf{u}) \int -\nabla_g E(\mu_E(g(\mathbf{u}))|\mathbf{x}) dp^*(\mathbf{x}) \right.$$
$$\left. - 2\sigma^2 \, b(\mathbf{u}) \int -\nabla_g E(\mu_E(g(\mathbf{u}))|\mathbf{x}) dG(\mathbf{x}) \right]$$
$$= \mathbb{E}_{\mathbf{u} \sim p(\mathbf{u})} \left[ 2\sigma^2 \, a(\mathbf{u}) \int -\nabla_g \frac{1}{Z} \exp\left( -\frac{1}{2\sigma^2} ||\mu_E(g(\mathbf{u})) - \mu_E(\mathbf{x})||^2 \right) dp^*(\mathbf{x}) \right.$$
$$\left. - 2\sigma^2 \, b(\mathbf{u}) \int -\nabla_g \frac{1}{Z} \exp\left( -\frac{1}{2\sigma^2} ||\mu_E(g(\mathbf{u})) - \mu_E(\mathbf{x})||^2 \right) dG(\mathbf{x}) \right]$$
$$= \mathbb{E}_{\mathbf{u} \sim p(\mathbf{u})} \left[ a(\mathbf{u}) \int \frac{1}{Z} \exp\left( -\frac{1}{2\sigma^2} ||\mu_E(g(\mathbf{u})) - \mu_E(\mathbf{x})||^2 \right) \nabla_g ||\mu_E(g(\mathbf{u})) - \mu_E(\mathbf{x})||^2 dp^*(\mathbf{x}) \right.$$
$$\left. - b(\mathbf{u}) \int \frac{1}{Z} \exp\left( -\frac{1}{2\sigma^2} ||\mu_E(g(\mathbf{u})) - \mu_E(\mathbf{x})||^2 \right) \nabla_g ||\mu_E(g(\mathbf{u})) - \mu_E(\mathbf{x})||^2 dG(\mathbf{x}) \right]$$
$$= \mathbb{E}_{\mathbf{u} \sim p(\mathbf{u})} \left[ a(\mathbf{u}) \int E(\mu_E(g(\mathbf{u}))|\mathbf{x}) \nabla_g ||\mu_E(g(\mathbf{u})) - \mu_E(\mathbf{x})||^2 dp^*(\mathbf{x}) \right.$$
$$\left. - b(\mathbf{u}) \int E(\mu_E(g(\mathbf{u}))|\mathbf{x}) \nabla_g ||\mu_E(g(\mathbf{u})) - \mu_E(\mathbf{x})||^2 dG(\mathbf{x}) \right]$$

$\square$

Similar to the result from Arjovsky & Bottou (2017), the gradient of the generator drives the generator's samples in the embedding space $\mu_E(g(\mathbf{u}))$ towards embeddings of the points from the dataset $\mu_E(\mathbf{x})$ weighted by their likelihood $E(\mu_E(g(\mathbf{u}))|\mathbf{x})$ under the real data. For an arbitrary encoder $E$, real and fake samples in the embedding may be far apart. As such, the coefficients $E(\mu_E(g(\mathbf{u}))|\mathbf{x})$ can be arbitrarily small, thereby resulting in vanishing gradients for the generator.

The second part of the theorem states that $C(I_c)$ is a continuous monotonic function, and $C(I_c) \to \delta > 0$ as $I_c \to 0$. This is the main result, and relies on the fact that $\mathrm{KL}[E(\mathbf{z}|\mathbf{x})||r(\mathbf{z})] \leq I_c$. The intuition behind this result is that, for any two inputs $\mathbf{x}$ and $\mathbf{y}$, their encoded distributions $E(\mathbf{z}|\mathbf{x})$ and $E(\mathbf{z}|\mathbf{y})$ have means that cannot be more than some distance apart, and that distance shrinks with $I_c$. This allows us to bound $E(\mu_E(\mathbf{y}))|\mathbf{x})$ below by $C(I_c)$, which ensures that the coefficients on the vectors in the theorem above are always at least as large as $C(I_c)$.

*Proof.* Let $r(\mathbf{z}) = \mathcal{N}(0, I)$ be the prior distribution and suppose the KL divergence for all $\mathbf{x}$ in the dataset and all $g(\mathbf{u})$ generated by the generator are bounded by $I_c$

$$\mathrm{KL}\left[E(\mathbf{z}|\mathbf{x})||r(\mathbf{z})\right] \leq I_c, \quad \forall \mathbf{x}, \ \mathbf{x} \sim p^*(\mathbf{x})$$
$$\mathrm{KL}\left[E(\mathbf{z}|g(\mathbf{u}))||r(\mathbf{z})\right] \leq I_c, \quad \forall \mathbf{u}, \ \mathbf{u} \sim p(\mathbf{u}).$$

From the definition of the KL-divergence we can bound the length of all embedding vectors,

$$\mathrm{KL}\left[E(\mathbf{z}|\mathbf{x})||r(\mathbf{z})\right] = \frac{1}{2}\left(K\sigma^2 + \mu_E(\mathbf{x})^T\mu_E(\mathbf{x}) - K - K\log\sigma^2\right) \leq I_c$$
$$||\mu_E(\mathbf{x})||^2 \leq 2I_c - K\sigma^2 + K + K\log\sigma^2,$$

and similarly for $||\mu_E(g(\mathbf{u}))||^2$, with $K$ denoting the dimension of $Z$. A lower bound on $E(\mu_E(g(\mathbf{u}))|\mathbf{x})$, where $\mathbf{u} \sim p(\mathbf{u})$ and $\mathbf{x} \sim p^*(\mathbf{x})$, can then be determined by

$$\log\left(E(\mu_E(g(\mathbf{u}))|\mathbf{x})\right) = -\frac{1}{2\sigma^2}\left(\mu_E(g(\mathbf{u})) - \mu_E(\mathbf{x})\right)^T\left(\mu_E(g(\mathbf{u})) - \mu_E(\mathbf{x})\right) - \frac{K}{2}\log\sigma^2 - \frac{K}{2}\log 2\pi$$

Since $||\mu_E(\mathbf{x})||^2, ||\mu_E(g(\mathbf{u}))||^2 \leq 2I_c - K\sigma^2 + K + K\log\sigma^2$,

$$||\mu_E(g(\mathbf{u})) - \mu_E(\mathbf{x})||^2 \leq 8I_c - 4K\sigma^2 + 4K + 4K\log\sigma^2,$$

and it follows that

$$-\frac{1}{2\sigma^2}\left(\mu_E(g(\mathbf{u})) - \mu_E(\mathbf{x})\right)^T\left(\mu_E(g(\mathbf{u})) - \mu_E(\mathbf{x})\right) \geq -4\sigma^{-2}I_c + 2K - 2K\sigma^{-2} - 2K\sigma^{-2}\log\sigma^{-2}.$$

The likelihood is therefore bounded below by

$$\log\left(E(\mu_E(g(\mathbf{u}))|\mathbf{x})\right) \geq -4\sigma^{-2}I_c + 2K - 2K\sigma^{-2} - 2K\sigma^{-2}\log\sigma^{-2} - \frac{K}{2}\log\sigma^2 - \frac{K}{2}\log 2\pi$$

Since $-\sigma^{-2} - \sigma^{-2}\log\sigma^{-2} \geq -1$,

$$\log\left(E(\mu_E(g(\mathbf{u}))|\mathbf{x})\right) \geq -4\sigma^{-2}I_c - \frac{K}{2}\log\sigma^2 - \frac{K}{2}\log 2\pi \tag{14}$$

From the KL constraint, we can derive a lower bound $\ell(I_c)$ and an upper bound $\mathcal{U}(I_c)$ on $\sigma^2$.

$$\frac{1}{2}\left(K\sigma^2 + \mu_E(\mathbf{x})^T\mu_E(\mathbf{x}) - K - K\log\sigma^2\right) \leq I_c$$
$$\sigma^2 - 1 - \log\sigma^2 \leq \frac{2I_c}{K}$$
$$\log\sigma^2 \geq -\frac{2I_c}{K} - 1$$
$$\sigma^2 \geq \exp\left(-\frac{2I_c}{K} - 1\right) = \ell(I_c)$$

For the upper bound, since $\sigma^2 - \log\sigma^2 > \frac{1}{2}\sigma^2$,

$$\sigma^2 - 1 - \log\sigma^2 \leq \frac{2I_c}{K}$$
$$\frac{1}{2}\sigma^2 - 1 < \frac{2I_c}{K}$$
$$\sigma^2 < \frac{4I_c}{K} + 2 = \mathcal{U}(I_c)$$

Substituting $\ell(I_c)$ and $\mathcal{U}(I_c)$ into Equation 14, we arrive at the following lower bound

$$E(\mu_E(g(\mathbf{u}))|\mathbf{x}) > \exp\left(-4I_c\exp\left(\frac{2I_c}{K} + 1\right) - \frac{K}{2}\log\left(\frac{4I_c}{K} + 2\right) - \frac{K}{2}\log 2\pi\right) = C(I_c).$$

$\square$

## B  GRADIENT PENALTY

To combine VDB with gradient penalty, we use the reparameterization trick to backprop through the encoder when computing the gradient of the discriminator with respect to the inputs.

$$
\begin{aligned}
J(D, E) = \min_{D,E} \quad & \mathbb{E}_{x \sim p^*(\mathbf{x})} \left[ \mathbb{E}_{\mathbf{z} \sim E(\mathbf{z}|\mathbf{x})} \left[ -\log\left(D(\mathbf{z})\right) \right] \right] + \mathbb{E}_{\mathbf{x} \sim G(\mathbf{x})} \left[ \mathbb{E}_{\mathbf{z} \sim E(\mathbf{z}|\mathbf{x})} \left[ -\log\left(1 - D(\mathbf{z})\right) \right] \right] \\
& + w_{GP} \mathbb{E}_{x \sim p^*(\mathbf{x})} \left[ \mathbb{E}_{\epsilon \sim \mathcal{N}(0,I)} \left[ \frac{1}{2} ||\nabla_x D(\mu_E(x) + \Sigma_E(x)\epsilon)||^2 \right] \right] \\
\text{s.t.} \quad & \mathbb{E}_{\mathbf{x} \sim \tilde{p}(\mathbf{x})} \left[ \text{KL} \left[ E(\mathbf{z}|\mathbf{x}) || r(\mathbf{z}) \right] \right] \leq I_c,
\end{aligned}
\tag{15}
$$

The coefficient $w_{GP}$ weights the gradient penalty in the objective, $w_{GP} = 10$ for the image generation, $w_{GP} = 1$ for motion imitation, and $w_{GP} = 0.1$ (C-maze) or $w_{GP} = 0.01$ (S-maze) for the IRL tasks. The gradient penalty is applied only to real samples $p^*(x)$. We have experimented with apply the penalty to both real and fake samples, but found that performance was worse than penalizing only gradients from real samples. This is consistent with the GP implementation from Mescheder et al. (2018).

## C  IMITATION LEARNING

**Experimental Setup:** The goal of the motion imitation tasks is to train a simulated agent to mimic a demonstration provided in the form of a mocap clip recorded from a human actor. We use a similar experimental setup as Peng et al. (2018), with a 34 degrees-of-freedom humanoid character. The state $\mathbf{s}$ consists of features that represent the configuration of the character's body (link positions and velocities). We also include a phase variable $\phi \in [0, 1]$ among the state features, which records the character's progress along the motion and helps to synchronize the character with the reference motion. With 0 and 1 denoting the start and end of the motion respectively. The action $\mathbf{a}$ sampled from the policy $\pi(\mathbf{a}|\mathbf{s})$ specifies target poses for PD controller positioned at each joint. Given a state, the policy specifies a Gaussian distribution over the action space $\pi(\mathbf{a}|\mathbf{s}) = \mathcal{N}(\mu(\mathbf{s}), \Sigma)$, with a state-dependent mean $\mu(\mathbf{s})$ and fixed diagonal covariance matrix $\Sigma$. $\mu(\mathbf{s})$ is modeled using a 3-layered fully-connected network with 1024 and 512 hidden units, followed by a linear output layer that specifies the mean of the Gaussian. ReLU activations are used for all hidden layers. The value function is modeled with a similar architecture but with a single linear output unit. The policy is queried at $30Hz$. Physics simulation is performed at 1.2kHz using the Bullet physics engine Bullet (2015).

Given the rewards from the discriminator, PPO (Schulman et al., 2017) is used to train the policy, with a stepsize of $2.5 \times 10^{-6}$ for the policy, a stepsize of $0.01$ for the value function, and a stepsize of $10^{-5}$ for the discriminator. Gradient descent with momentum 0.9 is used for all models. The PPO clipping threshold is set to 0.2. When evaluating the performance of the policies, each episode is simulated for a maximum horizon of $20s$. Early termination is triggered whenever the character's torso contacts the ground, leaving the policy is a maximum error of $\pi$ radians for all remaining timesteps.

**Phase-Functioned Discriminator:** Unlike the policy and value function, which are modeled with standard fully-connected networks, the discriminator is modeled by a phase-functioned neural network (PFNN) to explicitly model the time-dependency of the reference motion (Holden et al., 2017). While the parameters of a network are generally fixed, the parameters of a PFNN are functions of the phase variable $\phi$. The parameters $\theta$ of the network for a given $\phi$ is determined by a weighted combination of a set of fixed parameters $\{\theta_0, \theta_1, ..., \theta_k\}$,

$$
\theta = \sum_{i=0}^{k} w_i(\phi)\, \theta_i \,,
$$

where $w_i(\phi)$ is a phase-dependent weight for $\theta_i$. In our implementation, we use $k = 5$ sets of parameters and $w_i(\phi)$ is designed to linearly interpolate between two adjacent sets of parameters for each phase $\phi$, where each set of parameters $\theta_i$ corresponds to a discrete phase value $\phi_i$ spaced

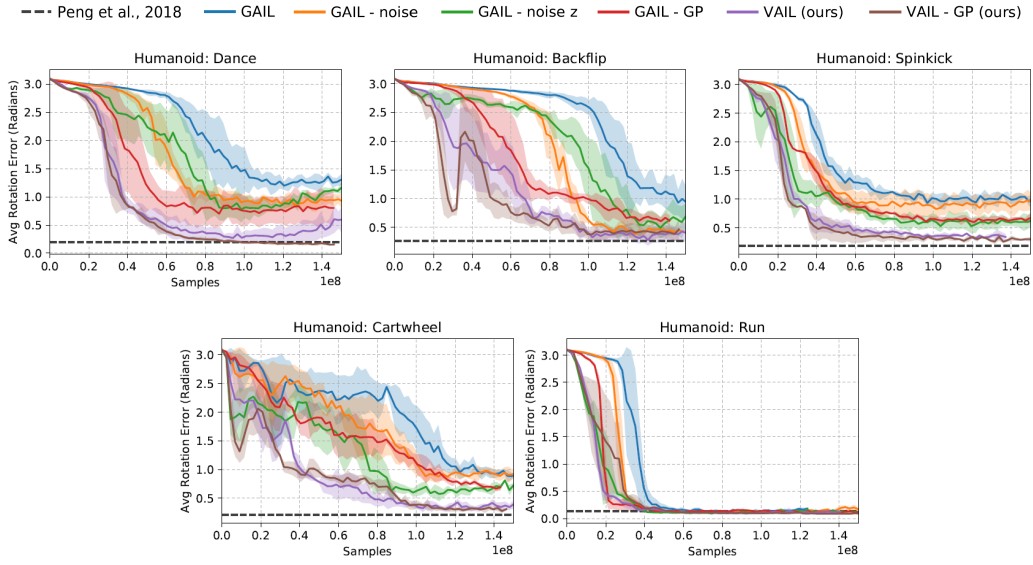

Figure 10: Learning curves comparing VAIL to other methods for motion imitation. Performance is measured using the average joint rotation error between the simulated character and the reference motion. Each method is evaluated with 3 random seeds.

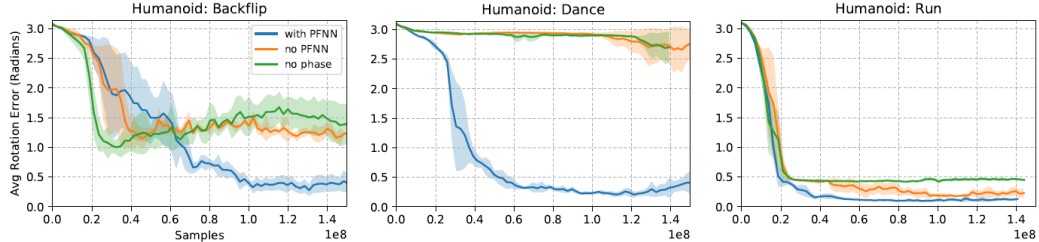

Figure 11: Learning curves comparing VAIL with a discriminator modeled by a phase-functioned neural network (PFNN), to modeling the discriminator with a fully-conneted network that receives the phase-variable $\phi$ as part of the input (no PFNN), and a discriminator modeled with a fully-connected network but does not receive $\phi$ as an input (no phase).

uniformly between $[0, 1]$. For a given value of $\phi$, the parameters of the discriminator are determined according to

$$\theta = w_i(\phi)\theta_i + w_{i+1}(\phi)\theta_{i+1}$$

where $\theta_i$ and $\theta_{i+1}$ correspond to the phase values $\phi_i \leq \phi < \phi_{i+1}$ that form the endpoints of the phase interval that contains $\phi$. A PFNN is used for all motion imitation experiments, both VAIL and GAIL, except for those that use the approach proposed by Merel et al. (2017), which use standard fully-connected networks for the discriminator. Figure 11 compares the performance of VAIL when the discriminator is modeled with a phase-functioned neural network (with PFNN) to discriminators modeled with standard fully-connected networks. We increased the size of the layers of the fully-connected nets to have a similar number of parameters as a PFNN. We evaluate the performance of fully-connected nets that receive the phase variable $\phi$ as part of the input (no PFNN), and fully-connected nets that do not receive $\phi$ as an input. The phase-functioned discriminator leads to significant performance improvements across all tasks evaluated. Policies trained without a phase variable performs worst overall, suggesting that phase information is critical for performance. All methods perform well on simpler skills, such as running, but the additional phase structure introduced by the PFNN proved to be vital for successful imitation of more complex skills, such as the dance and backflip.

Next we compare the accuracy of discriminators trained using different methods. Figure 12 illustrates accuracy of the discriminators over the course of training. Discriminators trained via GAIL quickly overpowers the policy, and learns to accurately differentiate between samples, even when

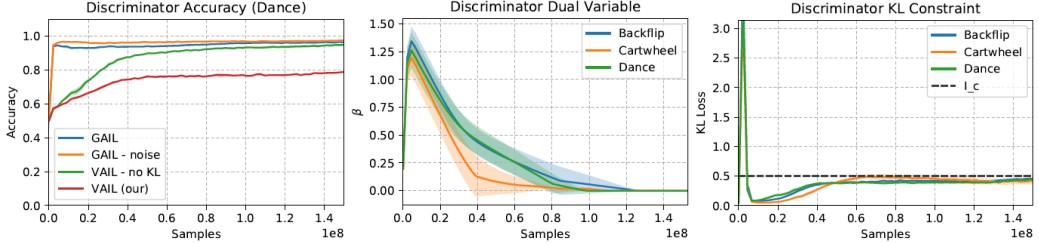

Figure 12: **Left:** Accuracy of the discriminator trained using different methods for imitating the dance skill. **Middle:**. Value of the dual variable $\beta$ over the course of training. **Right:** KL loss over the course of training. The dual gradient descent update for $\beta$ effectively enforces the VDB constraint $I_c$.

instance noise is applied to the inputs. VAIL without the KL constraint slows the discriminator's progress, but nonetheless reaches near perfect accuracy with a larger number of samples. Once the KL constraint is enforced, the information bottleneck constrains the performance of the discriminator, converging to approximately $80\%$ accuracy. Figure 12 also visualizes the value of $\beta$ over the course of training for motion imitation tasks, along with the loss of the KL term in the objective. The dual gradient descent update effectively enforces the VDB constraint $I_c$.

**Video Imitation:** In the video imitation tasks, we use a simplified 2D biped character in order to avoid issues that may arise due to depth ambiguity from monocular videos. The biped character has a total of 12 degrees-of-freedom, with similar state and action parameters as the humanoid. The video demonstrations are generated by rendering a reference motion into a sequence of video frames, which are then provided to the agent as a demonstration. The goal of the agent is to imitate the motion depicted in the video, without access to the original reference motion, and the reference motion is used only to evaluate performance.

# D    INVERSE REINFORCEMENT LEARNING

## D.1    EXPERIMENTAL SETUP

**Environments**    We evaluate on two maze tasks, as illustrated in Figure 13. The C-maze is taken from Fu et al. (2017): in this maze, the agent starts at a random point within a small fixed distance of the mean start position. The agent has a continuous, 2D action space which allows it to accelerate in the $x$ or $y$ directions, and is able to observe its $x$ and $y$ position, but not its velocity. The ground truth reward is $r_t = -d_t - 10^{-3}\|a_t\|^2$, where $d_t$ is the agent's distance to the goal, and $a_t$ is its action (this action penalty is assumed to be zero in Figure 13). Episodes terminate after 100 steps; for evaluation, we report the undiscounted mean sum of rewards over each episode The S-maze is larger variant of the same environment with an extra wall between the agent and its goal. To make the S-maze easier to solve for the expert, we added further reward shaping to encourage the agent to pass between the gaps between walls. We also increased the maximum control forces relative to the C-maze to enable more rapid exploration. Environments will be released along with the rest of our VAIRL implementations.

**Hyperparameters**    Policy networks for all methods were two-layer ReLU MLPs with 32 hidden units per layer. Reward and discriminator networks were similar, but with 32-unit mean and standard deviation layers inserted before the final layer for VDB methods. To generate expert demonstrations, we trained a TRPO (Schulman et al., 2015) agent on the ground truth reward for the training environment for 200 iterations, and saved 107 trajectories from each of the policies corresponding to the five final iterations. TRPO used a batch size of 10,000, a step size of 0.01, and entropy bonus with a coefficient of 0.1 to increase diversity. After generating demonstrations, we trained the IRL and imitation methods on a training maze for 200 iterations; again, our policy optimizer was TRPO with the same hyperparameters used to generate demonstrations. Between each policy update, we did 100 discriminator updates using Adam with a learning rate of $5 \times 10^{-5}$ and batch size of 32. For the C-maze our VAIRL runs used a target KL of $I_C = 0.5$, while for the more complex S-maze we

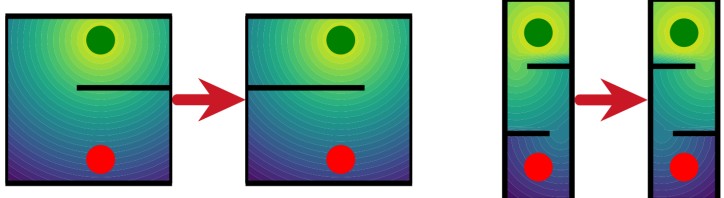

Figure 13: **Left:** The C-maze used for training and its mirror version used for testing. Colour contours show the ground truth reward function that we use to train the expert and evaluate transfer quality, while the red and green dots show the initial and goal positions, respectively. **Right:** The analogous diagram for the S-maze.

use a tighter target of $I_C = 0.05$. For the test C-maze, we trained new policies against the recovered reward using TRPO with the hyperparameters described above; for the test S-maze, we modified these parameters to use a batch size of 50,000 and learning rate of 0.001 for 400 iterations.

## D.2 RECOVERED REWARD FUNCTIONS

Figure 14 and 15 show the reward functions recovered by each IRL baseline on the C-maze and S-maze, respectively, along with sample trajectories for policies trained to optimize those rewards. Notice that VAIRL tends to recover smoother reward functions that match the ground truth reward more closely than the baselines. Addition of a gradient penalty enhances this effect for both AIRL and VAIRL. This is especially true in S-maze, where combining a gradient penalty with a variational discriminator bottleneck leads to a smooth reward that gradually increases as the agent nears its goal position at the top of the maze.

## E IMAGE GENERATION

We provide further experiment on image generation and details of the experimental setup.

### E.1 EXPERIMENTAL SETUP:

We use the non-saturating objective of Goodfellow et al. (2014) for all models except WGAN-GP. Following (Lucic et al., 2017), we compute FID on samples of size $10000^2$. We base our implementation on (Mescheder et al., 2018), where we do not use any batch normalization for both the generator and the discriminator. We use RMSprop (Hinton et al.) and a fixed learning rate for all experiments.

For convolutional GAN, variational discriminative bottleneck is implemented as a 1x1 convolution on the final embedding space that outputs a Gaussian distribution over $Z$ parametrized with a mean and a diagonal covariance matrix. For all image experiments, we preserve the dimensionality of the latent space. All experiments use adaptive $\beta$ update with a dual stepsize of $\alpha_\beta = 10^{-5}$. We will make our code public. Similarly to VGAN, instance noise Sønderby et al. (2016); Arjovsky & Bottou (2017) is added to the final embedding space of the discriminator right before applying the classifier. Instance noise can be interpreted as a non-adaptive VGAN without a information constraint.

**Architecture:** For CIFAR-10, we use a resnet-based architecture adapted from (Mescheder et al., 2018) detailed in Tables 2, 3, and 4. For CelebA and CelebAHQ, we use the same architecture used in (Mescheder et al., 2018).

---

[2]For computing the FID we use the public implementation of https://github.com/mseitzer/pytorch-fid.

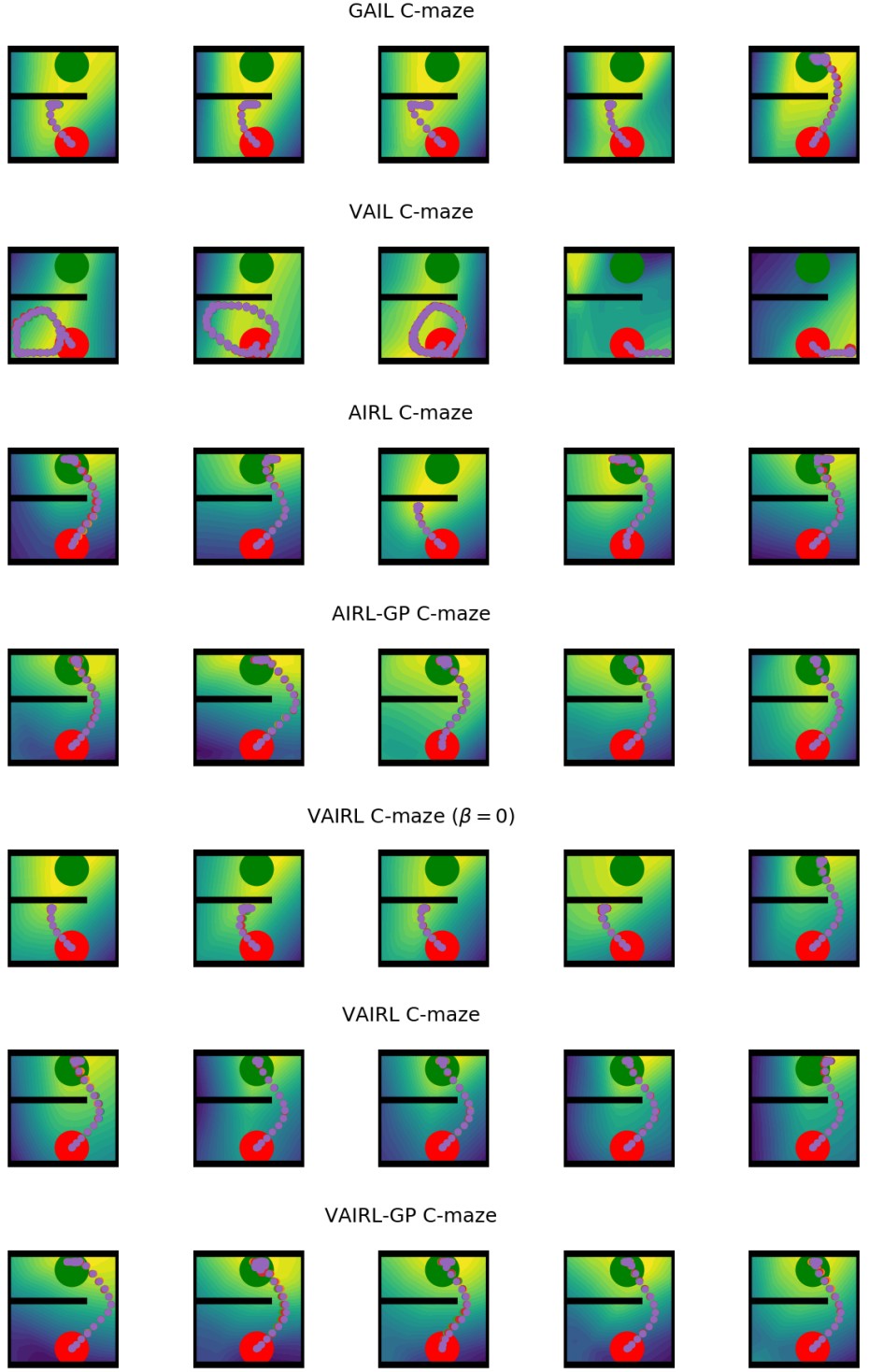

Figure 14: Visualizations of recovered reward functions transferred to the mirrored C-maze. Also shown are trajectories executed by policies trained to maximize the corresponding reward in the new environment.

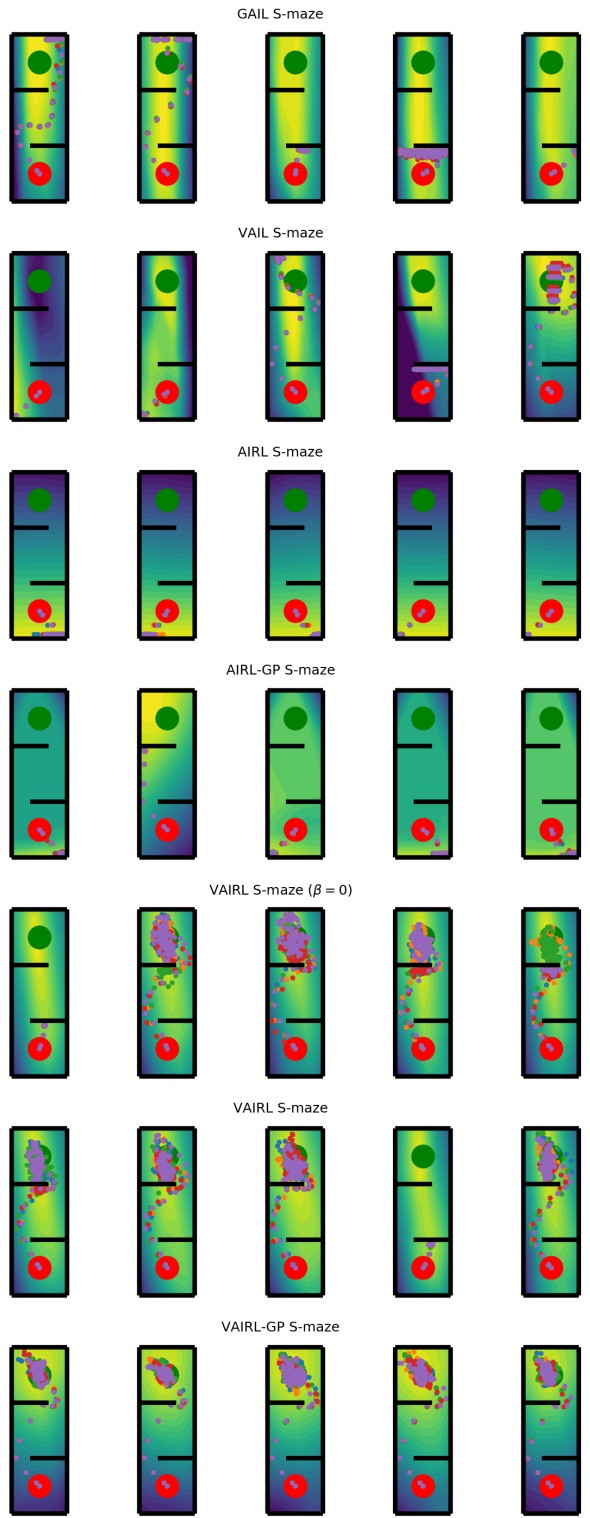

Figure 15: Visualizations of recovered reward functions transferred to the mirrored S-maze, like Figure 14.

| Layer | Output size | Filter |
|---|---|---|
| FC | $256 \cdot 4 \cdot 4$ | $256 \rightarrow 256 \cdot 4 \cdot 4$ |
| Reshape | $256 \times 4 \times 4$ | |
| Resnet-block | $128 \times 4 \times 4$ | $256 \rightarrow 128$ |
| Upsample | $128 \times 8 \times 8$ | |
| Resnet-block | $64 \times 8 \times 8$ | $128 \rightarrow 64$ |
| Upsample | $64 \times 16 \times 16$ | |
| Resnet-block | $32 \times 16 \times 16$ | $64 \rightarrow 32$ |
| Upsample | $32 \times 32 \times 32$ | |
| Resnet-block | $32 \times 32 \times 32$ | |
| Conv2D | $3 \times 32 \times 32$ | $32 \rightarrow 32$ |

Table 2: CIFAR-10 Generator

| Layer | Output size | Filter |
|---|---|---|
| Conv2D | $32 \times 32 \times 32$ | $3 \rightarrow 32$ |
| Resnet-block | $64 \times 32 \times 32$ | $32 \rightarrow 64$ |
| AvgPool2D | $64 \times 16 \times 16$ | |
| Resnet-block | $128 \times 16 \times 16$ | $64 \rightarrow 128$ |
| AvgPool2D | $128 \times 8 \times 8$ | |
| Resnet-block | $256 \times 8 \times 8$ | $128 \rightarrow 256$ |
| AvgPool2D | $256 \times 4 \times 4$ | |
| FC | $1$ | $256 \cdot 4 \cdot 4 \rightarrow 1$ |

Table 3: CIFAR-10 Discriminator

## E.2 RESULTS

**CIFAR-10:** We compare our approach with recent stabilization techniques: WGAN-GP (Gulrajani et al., 2017b), instance noise (Sønderby et al., 2016; Arjovsky & Bottou, 2017), spectral normalization (Miyato et al., 2018), and gradient penalty (Mescheder et al., 2018). We train report the networks at 750k iterations. We use $I_c = 0.1$, and a coefficient of $w_{GP} = 10$ for the gradient penalty, which is the same as the value used by the implementation from Mescheder et al. (2018). See Figure 16 for visual comparisons of randomly generated samples.

**CelebA:** On the CelebA (Liu et al., 2015) dataset, we generate images of size $128 \times 128$ with $I_c = 0.2$. On this dataset we do not see a big improvement upon the other baselines. This is likely because the architecture has been effectively tuned for this task, reflected by the fact that even the vanilla GAN trains fine on this dataset. All GAN, GP, and VGAN-GP obtain a similar FID scores of 7.64, 7.76, 7.25 respectively. See Figure 17 for more qualitative results with our approach.

**CelebAHQ:** VGAN can also be trained on on CelebAHQ Karras et al. (2018) at 1024 by 1024 resolution directly, without progressive growing (Karras et al., 2018). We use $I_c = 0.1$ and train with VGAN-GP. We train on a single Tesla V100, which fits a batch size of 8 in our experiments. Previous approaches (Karras et al., 2018; Mescheder et al., 2018) use a larger batch size and train over multiple GPUs. The model was trained for 300k iterations.

| Layer | Output size | Filter |
|---|---|---|
| Conv2D | $32 \times 32 \times 32$ | $3 \rightarrow 32$ |
| Resnet-block | $64 \times 32 \times 32$ | $32 \rightarrow 64$ |
| AvgPool2D | $64 \times 16 \times 16$ | |
| Resnet-block | $128 \times 16 \times 16$ | $64 \rightarrow 128$ |
| AvgPool2D | $128 \times 8 \times 8$ | |
| Resnet-block | $256 \times 8 \times 8$ | $128 \rightarrow 256$ |
| AvgPool2D | $256 \times 4 \times 4$ | |
| $1 \times 1$ Conv2D | $2 \cdot 256 \times 4 \times 4$ | $256 \rightarrow 2 \cdot 256$ |
| Sampling | $256 \times 4 \times 4$ | |
| FC | $1$ | $256 \cdot 4 \cdot 4 \rightarrow 1$ |

Table 4: CIFAR-10 Discriminator with VDB

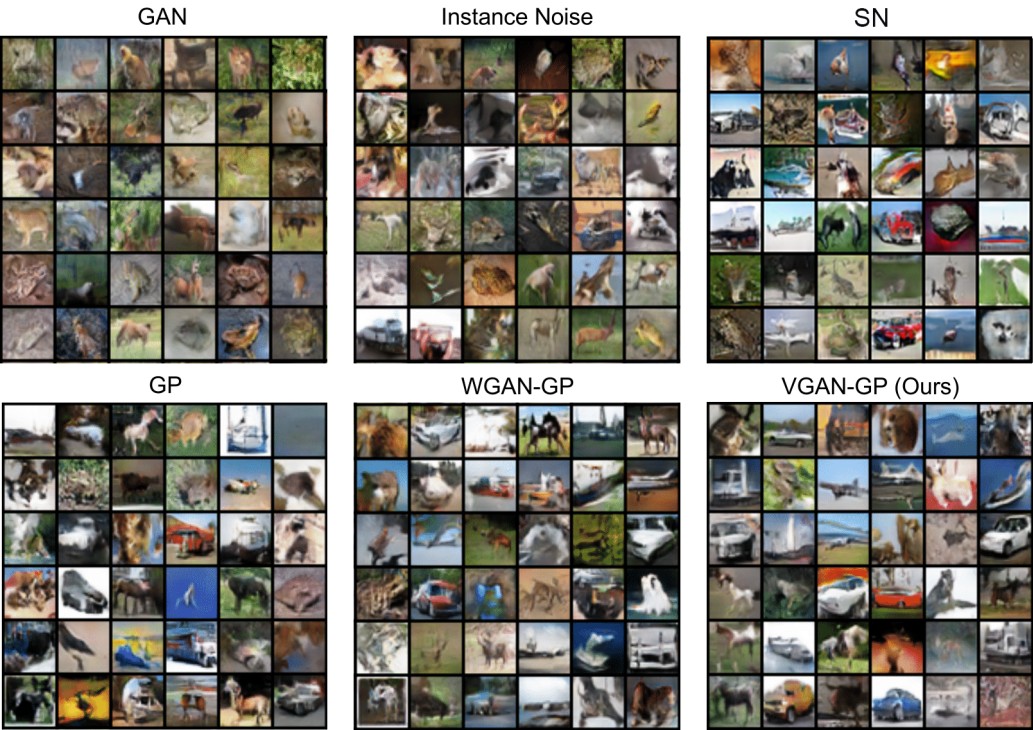

Figure 16: Random results on CIFAR-10 (Krizhevsky et al.): GAN (Goodfellow et al., 2014) FID: 63.6, instance noise (Sønderby et al., 2016; Arjovsky & Bottou, 2017) FID: 30.7, spectral normalization (SN) (Miyato et al., 2018) FID: 23.9, gradient penalty (GP) (Mescheder et al., 2018) FID: 22.6, WGAN-GP Gulrajani et al. (2017b) FID: 19.9, and the proposed VGAN-GP FID: 18.1. The samples produced by VGAN-GP (right) look the most realistic where objects like vehicles may be discerned.

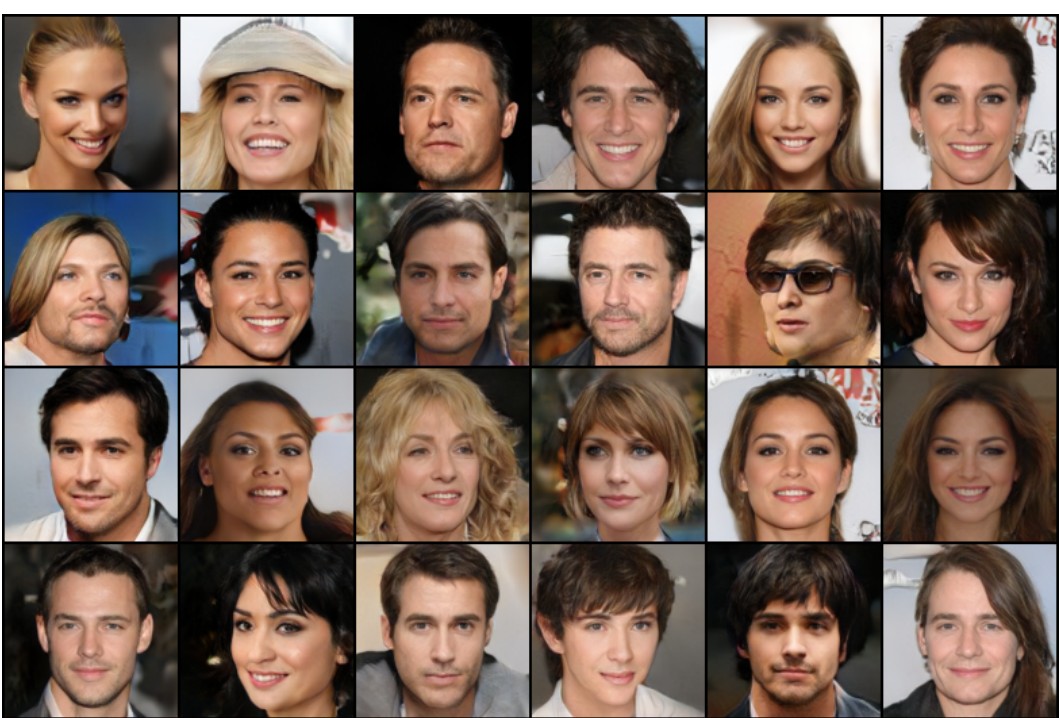

Figure 17: Random VGAN samples on CelebA 128 × 128 at 300k iterations.

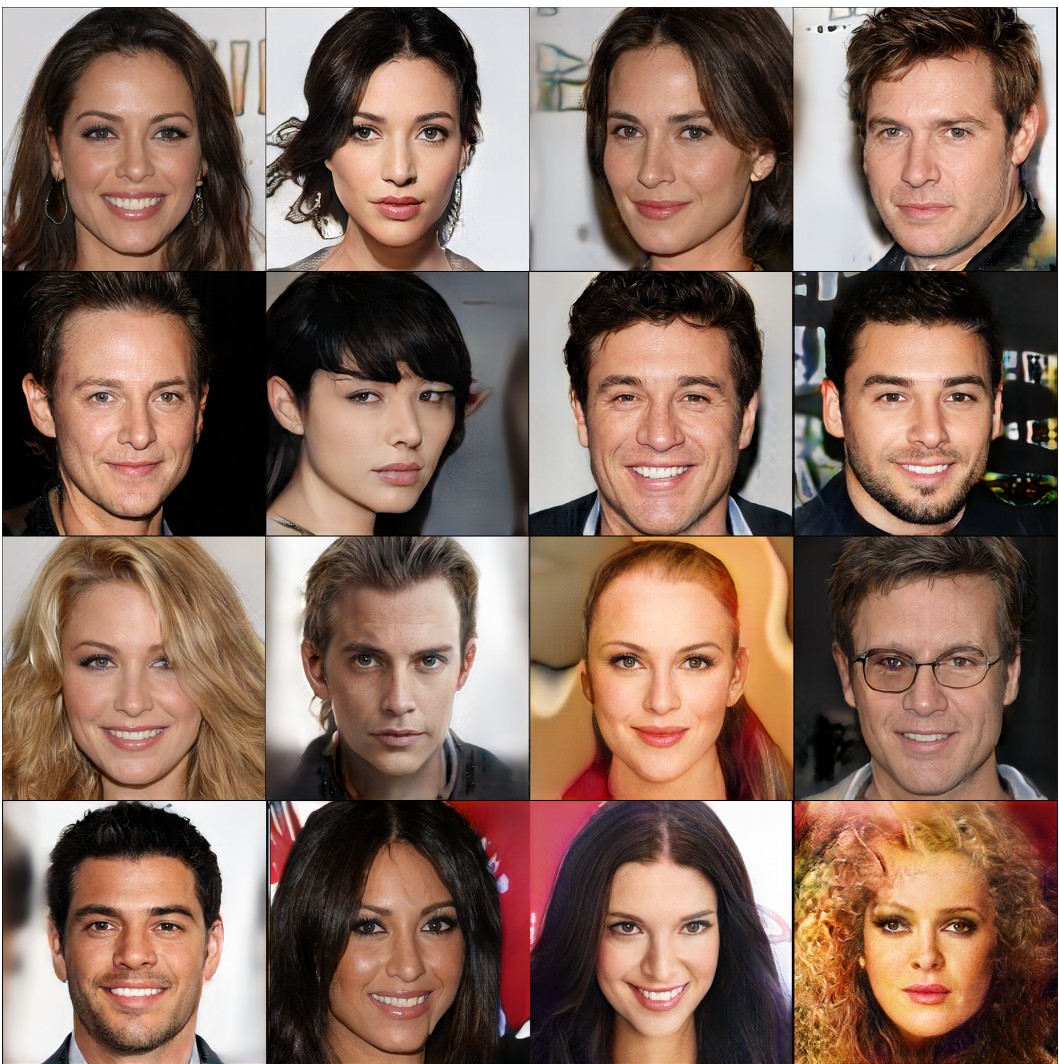

Figure 18: VGAN samples on CelebA HQ (Karras et al., 2018) 1024 × 1024 resolution at 300k iterations. Models are trained from scratch at full resolution, without the progressive scheme proposed by Karras et al. (2017).

