# OpenReview forum: "Variational Discriminator Bottleneck: Improving Imitation Learning, Inverse RL, and GANs by Constraining Information Flow"
_ICLR.cc/2019/Conference_

### Official Review · AnonReviewer1 · 2018-10-31
**Good showcase of the application and benefits of the VIB in GANs, minor corrections suggested.**

**Rating:** 8
**Confidence:** 3

**Review:**

Summary:
The authors propose to apply the Deep Variational Information Bottleneck (VIB) method of [1] on discriminator networks in various adversarial-learning-based scenarios. They propose a way to adaptively update the value for the bêta hyper-parameter to respect the constraint on I(X,Z). Their technique is shown to stabilize/allow training when P_g and P_data do not overlap, similarly to WGAN and gradient-penalty based approaches, by essentially pushing their representation distributions (p_z) to overlap with the mutual information bottleneck. It can also be considered as an adaptive version of instance noise, which serves the same goal. The method is evaluated on different adversarial learning setup (imitation learning, inverse reinforcement learning and GANs), where it compares positively to most related methods. Best results for ‘classical’ adversarial learning for image generation are however obtained when combining the proposed VIB with gradient penalty (which outperforms by itself the VGAN in this case).


Pros :
- This paper brings a good amount of evidence of the benefits to use the VIB formulation to adversarial learning by first showing the effect of such approach on a toy example, and then applying it to more complex scenarios, where it also boosts performance. The numerous experiments and analyses have great value and are a necessity as this paper mostly applies the VIB to new learning challenges.

- The proposition of a principled way of adaptively varying the value of Bêta to actually respect more closely the constraint I(X,Z) < I_c, which to my knowledge [1] does not perform, is definitely appealing and seems to work better than fixed Bêtas and does also bring the KL divergence to the desired I_c.

- The technique is fairly simple to implement and can be combined with other stabilization techniques such as gradient penalties on the discriminator.


Cons:

- In my view, the novelty of the approach is somewhat limited, as it seems like a straightforward application of the VIB from [1] for discriminators in adversarial learning, with the difference of using an adaptive Bêta.

- I think the Bêta-VAE [2] paper is definitely related to this paper and to the paper on which it is based [1] and should thus be cited as the authors use a similar regularization technique, albeit from a different perspective, that restricts I(X,Z) in an auto-encoding task.

- I think the content of batches used to regularize E(z|x) w.r.t. to the KL divergence should be clarified, as the description of p^tilde “being a mixture of the target distribution and the generator” (Section 4) can let the implementation details be ambiguous. I think batches containing samples from both distributions can cause problems as the expectation of the KL divergence on a batch can be low even if the samples from both distributions are projected into different parts of the manifold. This makes me think batches are separated? Either way, this should be more clearly stated in the text.

- The last results for  the ‘traditional’ GAN+VIB show that in this case, gradient penalty (GP) alone outperforms the proposed VGAN, and that both can be combined for best results. I thus wonder if the results in all other experiments could show similar trends if GP had been tested in these cases as well. In the imitation learning task, authors compare with instance noise, but not with GP, which for me are both related to VIB in what they try to accomplish. Was GP tested in Imitation Learning/Inverse RL ? Was it better? Could it still be combined with VIB for better results?

- In the saliency map of Figure 5, I’m unclear as to what the colors represent (especially on the GAIL side). I doubt that this is simply due to the colormap used, but this colormap should be presented.

Overall, I think this is an interesting and relevant paper that I am very likely to suggest to peers working on adversarial learning, and should therefore be presented. I think the limited novelty is counterbalanced by the quality of empirical analysis. Some clarity issues and missing citations should be easy to correct. I appreciate the comparison and combination with a competitive method (Gradient Penalty) in Section 5.3, but I wish similar results were present in the other experiments, in order to inform readers if, in these cases as well, combining VIB with GP leads to the best performance.

[1] Deep Variational Information Bottleneck, (Alemi et al. 2017)
[2] beta-VAE: Learning Basic Visual Concepts with a Constrained Variational Framework (Higgins et al. 2017)

---

> ### Author Response · Authors · 2018-11-15
> **Reply to AnonReviewer1**
>
> Thank you for the insight and feedback. We have included additional experiments to further compare with previous techniques, along with some additional clarifications.
>
> Re: additional citations
> Thank you for the pointers, we have included the additional citations.
>
> Re: GP for other task
> We have conducted additional motion imitation experiments with GAIL - GP and VAIL - GP [Figure 4, Table 1]. We also added experiments incorporating GP for the inverse RL tasks [Figure 7]. As in image generation, GP does indeed significantly improve the performance of GAIL. However, VAIL still performs better on most of the tasks, and VAIL - GP achieves the best performance overall.
>
> Re: content of batches used to compute KL divergence
> We have added additional information to the paper to clarify the content of each batch [Section 4 above equation 11]. Each batch of data used to compute the expected KL contains an equal number of real and fake samples. The encoder maps each input sample to an individual distribution in Z. The KL divergence is computed separately for the distribution of each input, and then averaged across the batch, as opposed to computing the KL divergence across samples within a batch. Therefore, if the real and fake distributions are mapped to different parts of the manifold, it should result in a large KL.
>
> Re: saliency maps
> We have added a colormap to Figure 5. The colors on the saliency map represent the magnitude of the discriminator’s gradient with respect to each pixel and color channel in the input image. The gradients are visualized for each color channel, which results in the different colors. The same procedure is used to compute the gradients for GAIL.

---

### Official Review · AnonReviewer2 · 2018-11-02
**Inovative technique, Impressive results**

**Rating:** 10
**Confidence:** 4

**Review:**

The paper "Variational Discriminator Bottleneck: Improving Imitation Learning, Inverse RL, and GANs by Constraining Information Flow" tackles the problem of discriminator over-fitting in adversarial learning. Balancing the generator and the discriminator is difficult in generative adversarial techniques, as a too good discriminator prevents the generator to converge toward effective distributions. The idea is to introduce an information constraint on a intermediate layer, called information bottleneck, which limits the content of this layer to the most discriminative features of the input. Based on this limited representation of the input, the disciminator is constrained to longer tailed-distributions, maintaining some uncertainty on simulated data distributions. Results show that the proposal outperforms previous researches on discriminator over-fitting, such as noise adding in the discriminator inputs.

While the use of information bottleneck is not novel, its application in adversarial learning looks inovative and the results are impressive in a broad range of applications. The paper is well-written and easy to follow, though I find that it would be nice to give more insights on the intuition about information bottleneck in the preliminary section to make the paper self-contained (I had to read the previous work from Alemi et al (2016) to realize what information bottleneck can bring). My only question is about the setting of the constaint Ic: wouldn't it be possible to consider an adaptative version which could consider the amount of zeros gradients returned to the generator ?

---

> ### Author Response · Authors · 2018-11-15
> **Reply to AnonReviewer2**
>
> Thank you for the insight and feedback, we have included new experiments in the paper, along with some additional clarifications.
>
> Re: Adapt beta based on gradient magnitudes
> Yes, it might be possible to formulate a similar constraint for adaptively updating beta according to the gradient magnitudes. A constraint on the gradient norm can be added, then a Lagrangian can be constructed in a similar manner to yield an adaptive update for beta.

---

### Official Review · AnonReviewer3 · 2018-11-02
**a constraint on the discriminator of GAN model to maintain informative gradients**

**Rating:** 6
**Confidence:** 3

**Review:**

This paper proposed a constraint on the discriminator of GAN model to maintain informative gradients. It is completed by control the mutual information between the observations and the discriminator’s internal representation to be no bigger than a predefined value.  The idea is interesting and the discussions of applications in different areas are useful. However, I still have some concerns about the work:
1.	in the experiments about image generation, it seems that the proposed method does not enhance the performance obviously when compared to GP and WGAN-GP, Why the combination of VGAN and GP can enhance the performance greatly(How do they complementary to each other), what about the performance when combine VGAN with WGAN-GP?
2.	How do you combine VGAN and GP, is there any parameter to balance their effect?
3.	The author stated on page 2 that “the  proposed information bottleneck encourages the discriminator to ignore irrelevant cues, which then allows the generator to focus on improving the most discerning differences between real and fake samples”, a proof on theory or experiments should be used to illustrate this state.
4.	Is it possible to apply GP and WGAN-GP to the Motion imitation or adversarial inverse reinforcement learning problems? If so, will it perform better than VGAN?
5.	How about VGAN compares with Spectral norm GAN?

---

> ### Author Response · Authors · 2018-11-15
> **Reply to AnonReviewer3**
>
> Thank you for the insight and suggestions. We have added additional experiments and clarifications to the paper that aim to address each of your concerns -- we would really appreciate it if you could revisit your review in light of these additions and clarifications.
>
> Re: GP for other tasks
> We have conducted additional motion imitation experiments with GAIL - GP and VAIL - GP [Figure 4, Table 1]. We also added experiments incorporating GP for the inverse RL tasks [Figure 7]. As in image generation, GP does indeed significantly improve the performance of GAIL. However, VAIL still performs better on most of the tasks, and VAIL - GP achieves the best performance overall.
>
> Re: How are VGAN and GP combined
> We have added an additional section [Appendix B] that provides more information on how VDB and GP is combined. We use the reparameterization trick, as is done in VAEs, to backprop through the encoder to compute the gradient of the discriminator with respect to the inputs. There is a manually specified coefficient that weights the GP term in the objective, and we use the same value for the coefficient as [Mescheder et al., 2018] for image generation.
>
> Re: Combining VGAN and GP enhances performance
> The VDB and GP are complementary techniques since the VDB helps to prevent vanishing gradients and GP prevents exploding gradients. Therefore both methods regularize the gradients, but under different criteria.
>
> Re: Spectral norm
> We have included additional image generation experiments with spectral normalization [Figure 8]. Spectral normalization does show significant improvement over the vanilla GAN on CIFAR-10 (FID: 23.9), but our method still achieves a better score (FID: 18.1). The original spectral normalization paper [Miyato et al., 2018] reported an FID of 21.7 on CIFAR-10.

---

### Public Comment · (anonymous) · 2018-11-11
**GAN experiments writing indicating incorrect interpretations?**

"CelebAHQ: VGAN can also be trained on on CelebAHQ Karras et al. (2018) at 1024 by 1024 resolution directly, without progressive growing (Karras et al., 2018). We use Ic = 0.1 and train with VGAN-GP. We train on a single Tesla V100, which fits a batch size of 8 in our experiments. Previous approaches (Karras et al., 2018; Mescheder et al., 2018) use a larger batch size and train over multiple GPUs. While previous approaches have trained this for 300k iterations or more, our results are shown at 100k iterations."

Even though the authors don't intend to, this statement is likely to be misinterpreted that VGAN is the first GAN paper to show high resolution GAN samples without progressive growing of resolution or large batch sizes.

The batch size used in Mescheder et al is 24 while the authors use 8. Why would you call 24 "large" and 8 "small"? Secondly, 100k iterations is sufficient to start seeing good samples with most GAN architectures when the architecture uses residual connections and more iterations are needed to get more modes and sharper samples. You have shown a total of 8 samples. It is hard to say whether or not they were carefully picked.

As evidence for why this is likely to be misleading, I am quoting a comment from reddit: "Also of note: training 1024px image GANs without extremely large minibatches, progressive growing, or self-attention, just a fairly vanilla-sounding CNN and their discriminator penalization." Not providing the link because that breaks the anonymity of the paper.

Neither is it claimed or shown by the authors that Mescheder et al's model wouldn't produce good samples with a lower batch size or fewer (100K) iterations. The benefit to get it working for large resolution comes from the careful architecture designed by Mescheder et al and not from the bottleneck.

Two more issues with the claims made in the CIFAR-10 FID metrics section: (a) "VGAN is competitive with WGAN-GP and GP": The gap between VGAN and WGAN-GP is higher than WGAN-GP and VGAN-GP.  But the improvement over WGAN-GP is considered "significant" whereas the other gap is considered "competitive"?  (b) Is there any reason to show the metrics at the end of 750K iterations specifically? The plot shows that WGAN-GP training curve has a bigger negative slope at the cutoff point (750k) while VGAN-GP has flattened by then. It is worth showing the readers what happens when you train even a bit more, ie 1 million iterations when the difference isn't even that significant. Even though "VDB and GP are complementary techniques" morally, empirical conclusions may often not turn out to be the case.

---

> ### Author Response · Authors · 2018-11-11
> **Clarification**
>
> Thank you for your comment.
>
> The authors of the paper are not active on reddit and we do not have control over what reddit users post about our paper.
>
> We used a batch size of 8 in our work, and we mention this in the paper for completeness, and since this is a bit different from Meschederer et al., who used a batch size of 24 with 4 GPUs. We do not state that the batch size from Meschederer et al. is “extremely large” in our paper, we state that it is "larger" than 8, which is factually true (it’s not clear how to state this in any other way…). We did not claim that the smaller batch size of 8 is a contribution of our work, and we did not claim that our paper is the first to train high-resolution GANs without progressive growing of resolution. We do have results for a network trained for 300k iterations and we will add these results to the paper.
>
> We will refine the wording for the image generation experiments to further avoid these misinterpretations.

---

> > ### Public Comment · (anonymous) · 2018-11-12
> > **Response to Clarification**
> >
> > Thanks for your response clarifying one part of the comment.
> >
> > With respect to all the "We never claimed ...", the writing did not have factually false claims. However, isn't it normal to interpret that a statement like "previous approaches used larger batch sizes and multiple GPUs and our approach did not" is intended to "sound" as a contribution in comparison to prior work? 24 is larger than 8. 256 is also larger than 8. 2048 is also larger than 8. But it's not the same "larger". One is doable with a single V100. Another is doable with 32 V100s. Third is doable only on TPU. Wouldn't it make sense to say "We used smaller batch size (8 instead of 24 as in Mescheder et al) on a single V100 and trained for fewer iterations because of resource constraints. We also generate at full resolution directly as in Mescheder et al instead of progressive growing done in Karras et al"? Thanks for agreeing to refine the writing.

---

### Meta-Review · Area_Chair1 · 2018-12-14
**Intuitive idea that leads to impressive results!**

**Confidence:** 5
**Recommendation:** Accept (Poster)

**Metareview:**

The paper proposes a simple and general technique based on the information bottleneck to constrain the information flow in the discriminator of adversarial models. It helps to train by maintaining informative gradients. While the information bottleneck is not novel, its application in adversarial learning to my knowledge is, and the empirical evaluation demonstrates impressive performance on a broad range of applications. Therefore, the paper should clearly be accepted.